# Concept-based Adversarial Attack: a Probabilistic Perspective

**Andi Zhang**[*]
University of Warwick
University of Manchester
az381@cantab.ac.uk

**Xuan Ding**
The Chinese University of Hong Kong, Shenzhen
202322011119@mail.bnu.edu.cn

**Steven McDonagh**
University of Edinburgh
s.mcdonagh@ed.ac.uk

**Samuel Kaski**
University of Manchester
University of Aalto
samuel.kaski@aalto.fi

## Abstract

We propose a concept-based adversarial attack framework that extends beyond single-image perturbations by adopting a probabilistic perspective. Rather than modifying a single image, our method operates on an entire concept — represented by a distribution — to generate diverse adversarial examples. Preserving the concept is essential, as it ensures that the resulting adversarial images remain identifiable as instances of the original underlying category or identity. By sampling from this concept-based adversarial distribution, we generate images that maintain the original concept but vary in pose, viewpoint, or background, thereby misleading the classifier. Mathematically, this framework remains consistent with traditional adversarial attacks in a principled manner. Our theoretical and empirical results demonstrate that concept-based adversarial attacks yield more diverse adversarial examples and effectively preserve the underlying concept, while achieving higher attack efficiency. Code and examples can be found at https://github.com/andiac/ConceptAdv.

## 1 Introduction

Adversarial attacks aim to deceive a classifier while preserving the original meaning of the input object (Dalvi et al., 2004; Lowd & Meek, 2005a;b; Biggio & Roli, 2018). We refer to the manipulated instance as an adversarial example. Early work by Szegedy et al. (2014) and Goodfellow et al. (2015) introduced adversarial attacks against deep learning models for images.

In the image-based adversarial setting, it is widely accepted that controlling the geometric distance between the adversarial example and its original image is crucial for maintaining the original image's meaning. Consequently, many adversarial attack algorithms constrain perturbations using norms such as $L_1$, $L_2$, or $L_\infty$. Moreover, given the rapid progress in machine learning research, fair comparisons across different adversarial methods have become essential. Numerous benchmarks and competitions (Madry et al., 2018; Croce et al., 2020; Dong et al., 2021) therefore focus on attack success rates under the constraint that geometric distance does not exceed a threshold $\delta$.

However, as adversarial defense techniques improve, small geometric perturbations alone increasingly fail to generate adversarial examples that reliably fool classifiers, particularly when strong transferability is required (Song et al., 2018; Xiao et al., 2018b; Bhattad et al., 2020). This shortcoming has led researchers to explore unrestricted adversarial attacks, which involve larger geometric perturbations. Although "unrestricted" implies that adversarial examples need not be bounded by strict geometric distance, these examples must still remain faithful to the semantics of the original image; otherwise, the core goal of preserving the input's meaning is lost.

---

[*]Corresponding Author.

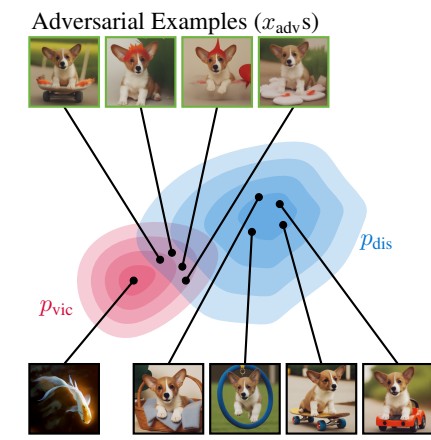

Figure 1: Comparison of a single-image adversarial attack (left) versus our proposed concept-based adversarial attack (right). In both cases, adversarial examples $x_{\text{adv}}$ are drawn from the product of a distance distribution $p_{\text{dis}}$ and a victim distribution $p_{\text{vic}}$. On the left, $p_{\text{dis}}$ is centered on a single image $x_{\text{ori}}$, so its overlap with $p_{\text{vic}}$ is small. Consequently, adversarial examples that successfully fool the victim classifier typically lose the original image's meaning, whereas those that preserve the original meaning fail to deceive the classifier. In contrast, on the right, $p_{\text{dis}}$ spans the original concept $\mathcal{C}_{\text{ori}}$, greatly increasing overlap with $p_{\text{vic}}$. As a result, the generated adversarial examples both maintain the concept's meaning and easily deceive the classifier. (A **green** image border indicates an example that successfully fools the classifier; **red** indicates failure.)

Zhang et al. (2024b) introduced a probabilistic perspective on adversarial attacks, demonstrating that traditional geometric constraints can be interpreted as specific "distance" distributions $p_{\text{dis}}$. Under this view, generating adversarial examples amounts to sampling from the product of $p_{\text{dis}}$ and the "victim" distribution $p_{\text{vic}}$, which represents the target classifier under attack. Importantly, $p_{\text{dis}}$ need not be induced solely by geometric distance. Instead, one can fit a probabilistic generative model (PGM) around the original image, allowing the PGM's semantic representation to implicitly define a semantics-based notion of distance. As illustrated on the left side of Figure 1, Zhang et al. (2024b) indicate that $p_{\text{dis}}$ should be centered on the original image.

Building on Zhang et al. (2024b)'s probabilistic perspective, we expand the distance distribution $p_{\text{dis}}$ from operating on a single image to operating on an entire concept, which can be represented by a probability distribution over images that correspond to the same underlying object, identity or category. As shown on the right side of Figure 1, this generalization introduces a new class of adversarial attacks. Rather than perturbing a single image, we generate a fresh image that captures the same underlying concept yet deceives the classifier. We refer to this approach as a concept-based adversarial attack. Mathematically, it remains consistent with traditional adversarial attacks when viewed under the probabilistic framework. As we demonstrate, broadening the distance distribution to concept-level information reduces its gap from the victim distribution $p_{\text{vic}}$, resulting in substantially higher attack success rates.

Our main contributions are as follows:

- **Concept-based adversarial attack**: We introduce a new type of adversarial attack that moves beyond single-image perturbations to a concept represented by a distribution, this new approach aligns with traditional adversarial attacks in a principled manner.

- **Concept augmentation**: We propose a practical concept augmentation strategy using modern generative models, enhancing the diversity of the distance distribution.

- **Theoretical and empirical validation**: We provide both theoretical proof and experimental evidence showing that expanding the attack from a single image to an entire concept reduces the distance between $p_{\text{vic}}$ and $p_{\text{dis}}$, boosting the attack efficiency.

- **Higher success rates**: Our experiments confirm that concept-based adversarial attacks achieve higher targeted attack success rates while preserving the original concept.

- **Practical Guidelines and Scenarios**: We provide practical guidelines and example application scenarios, detailed in Appendix K.

## 2 PRELIMINARIES

### 2.1 PROBABILISTIC GENERATIVE MODELS (PGMS) AND THEIR LIKELIHOODS

The goal of probabilistic generative models is to learn a parameterized distribution $p_\theta$ that approximates the true distribution $p$. In practice, we only observe a finite dataset $\mathcal{D} = \{x_1, \ldots, x_n\}$, and training is typically done by maximizing its likelihood. For image modeling, popular approaches such as VAEs (Kingma et al., 2013) and diffusion models (Song & Ermon, 2019; Ho et al., 2020) optimize a lower bound on the log-likelihood (the ELBO) rather than the likelihood itself. Thus, likelihood estimation in practice amounts to computing this ELBO (Burda et al., 2015; Nalisnick et al., 2019).

### 2.2 ADVERSARIAL ATTACK

Given a classifier $C : [0,1]^n \to \mathcal{Y}$, where $n$ is the input dimension and $\mathcal{Y}$ is the label space, consider an original image $x_{\mathrm{ori}} \in [0,1]^n$ and a target label $y_{\mathrm{tar}} \in \mathcal{Y}$. The goal of a targeted adversarial attack is to construct an adversarial example $x_{\mathrm{adv}}$ such that $C(x_{\mathrm{adv}}) = y_{\mathrm{tar}}$ while keeping $x_{\mathrm{adv}}$ close to $x_{\mathrm{ori}}$. The corresponding optimization problem is

$$\min \mathcal{D}(x_{\mathrm{ori}}, x_{\mathrm{adv}}) \quad \text{subject to} \quad C(x_{\mathrm{adv}}) = y_{\mathrm{tar}} \quad \text{and} \quad x_{\mathrm{adv}} \in [0,1]^n,$$

where $\mathcal{D}$ measures the distance (similarity) between $x_{\mathrm{ori}}$ and $x_{\mathrm{adv}}$, typically via an $\mathcal{L}_1$, $\mathcal{L}_2$, or $\mathcal{L}_\infty$ norm. Directly solving this constrained optimization can be challenging. To address this, Szegedy et al. (2014) propose a relaxation:

$$\min \ \mathcal{D}(x_{\mathrm{ori}}, x_{\mathrm{adv}}) \ + \ c \, f(x_{\mathrm{adv}}, y_{\mathrm{tar}}) \quad \text{subject to} \quad x_{\mathrm{adv}} \in [0,1]^n, \tag{1}$$

where $c$ is a constant, and $f$ is an objective function that guides the classifier's predictions toward the target label. In Szegedy et al. (2014)'s work, $f$ is taken to be the cross-entropy loss; Carlini & Wagner (2017) present additional choices for $f$.

### 2.3 PROBABILISTIC ADVERSARIAL ATTACK

By employing Langevin Dynamics as an optimizer for equation 1, Zhang et al. (2024b) derive a probabilistic perspective on adversarial attacks. They introduce the adversarial distribution:

$$p_{\mathrm{adv}}(x_{\mathrm{adv}} \mid x_{\mathrm{ori}}, y_{\mathrm{tar}}) \propto p_{\mathrm{vic}}(x_{\mathrm{adv}} \mid y_{\mathrm{tar}}) \, p_{\mathrm{dis}}(x_{\mathrm{adv}} \mid x_{\mathrm{ori}}), \tag{2}$$

where $p_{\mathrm{vic}}(x_{\mathrm{adv}} \mid y_{\mathrm{tar}}) \propto \exp\big(-c\, f(x_{\mathrm{adv}}, y_{\mathrm{tar}})\big)$ is the "victim" distribution emphasizing misclassification toward $y_{\mathrm{tar}}$, and $p_{\mathrm{dis}}(x_{\mathrm{adv}} \mid x_{\mathrm{ori}}) \propto \exp\big(-\mathcal{D}(x_{\mathrm{ori}}, x_{\mathrm{adv}})\big)$ is the "distance" distribution around $x_{\mathrm{ori}}$. This formulation leverages the fact that Langevin Dynamics converges to the corresponding Gibbs distribution (Lamperski, 2021), thereby providing a probabilistic interpretation of adversarial attack generation.

This probabilistic perspective aligns with traditional geometry-based adversarial attacks. For example, if $\mathcal{D}$ is the $\mathcal{L}_1$ norm, then $p_{\mathrm{dis}}(x_{\mathrm{adv}} \mid x_{\mathrm{ori}}) \propto \exp(-\|x_{\mathrm{adv}} - x_{\mathrm{ori}}\|_1)$ takes the form of a Laplace distribution. Similarly, if $\mathcal{D}$ is the squared $\mathcal{L}_2$ norm, then $p_{\mathrm{dis}}(x_{\mathrm{adv}} \mid x_{\mathrm{ori}}) \propto \exp(-\|x_{\mathrm{adv}} - x_{\mathrm{ori}}\|_2^2)$ is a Gaussian distribution.

Zhang et al. (2024b) indicate that the distance distribution $p_{\mathrm{dis}}$ can be any distribution centered around $x_{\mathrm{ori}}$, meaning the choice of $p_{\mathrm{dis}}$ implicitly defines the distance $\mathcal{D}$. Consequently, using a PGM centered on $x_{\mathrm{ori}}$ as $p_{\mathrm{dis}}$ yields a semantic-aware notion of distance. By then sampling from the corresponding adversarial distribution $p_{\mathrm{adv}}$, one can generate semantic-aware adversarial examples.

# 3 CONCEPT-BASED ADVERSARIAL ATTACK

## 3.1 CONCEPT DISTRIBUTION

We aim to extend adversarial attacks from operating on a single original image to operating on an original concept $\mathcal{C}_{\text{ori}}$. A concept is inherently abstract and subjective: it may refer to a specific physical object (e.g., a rubber duck), a particular identity such as the long-eared corgi puppy with a lighter left cheek shown in Figure 1, or a broader class such as "corgi," regardless of age, size, or specific attributes. **Although defining a concept in an absolute sense is difficult (Poeta et al., 2023), we can represent it through a concept distribution**, denoted by $p(\cdot \mid \mathcal{C}_{\text{ori}})$.

This distribution serves as an interface through which users can specify what the concept is. In practice, we recommend two ways for users to instantiate their notion of a concept:

- **Direct specification of a concept distribution**: The user may already possess a generative model or any other mechanism that directly provides a distribution $p(\cdot \mid \mathcal{C}_{\text{ori}})$ representing the concept.

- **Constructing the distribution from an image set**: The user may collect a set of images depicting the desired concept (e.g., different poses of the same corgi in Figure 1), and then train or fine-tune a probabilistic generative model (PGM) on this set to obtain the corresponding concept distribution $p(\cdot \mid \mathcal{C}_{\text{ori}})$. Here, $\mathcal{C}_{\text{ori}}$ is a set of images $\mathcal{C}_{\text{ori}} = \{x_{\text{ori}}^{(1)}, \ldots, x_{\text{ori}}^{(K)}\}$, where $K$ is the number of images depicting $\mathcal{C}_{\text{ori}}$.

In the remainder of this paper, we demonstrate the second approach, as it allows us to clearly showcase how concept-level information can be incorporated into adversarial attacks using accessible image data and standard generative modeling pipelines.

## 3.2 CONCEPT DISTRIBUTION AS A DISTANCE DISTRIBUTION

Building on the probabilistic perspective of adversarial attacks (Zhang et al., 2024b), a distribution used as a distance distribution can implicitly define a notion of distance. Therefore, by using the concept distribution defined in the previous subsection as the distance distribution, we implicitly define the distance between an adversarial example and the underlying concept. Formally,

$$p_{\text{adv}}(x_{\text{adv}} \mid \mathcal{C}_{\text{ori}}, y_{\text{tar}}) \propto p_{\text{vic}}(x_{\text{adv}} \mid y_{\text{tar}}) \, p_{\text{dis}}(x_{\text{adv}} \mid \mathcal{C}_{\text{ori}}) \tag{3}$$

where $p_{\text{adv}}(\cdot \mid \mathcal{C}_{\text{ori}}, y_{\text{tar}})$ is the adversarial distribution relative to the concept $\mathcal{C}_{\text{ori}}$ and the target label $y_{\text{tar}}$. The distance distribution $p_{\text{dis}}(\cdot \mid \mathcal{C}_{\text{ori}})$ is the concept distribution.

Comparing (2) and (3) shows that the only modification is replacing $x_{\text{ori}}$ with $\mathcal{C}_{\text{ori}}$. Hence, the probabilistic adversarial attack (Zhang et al., 2024b) is the special case $\mathcal{C}_{\text{ori}} = \{x_{\text{ori}}\}$ (i.e., $|\mathcal{C}_{\text{ori}}| = 1$). This straightforward and compact expansion allows us to heavily reuse the implementation of the probabilistic adversarial attack, making probabilistic adversarial attack (ProbAttack[1]) a natural ablation baseline for our method.

While intuition suggests that expanding the perturbation space should produce stronger adversarial examples, rigorous justification is needed. In the following sections, we adopt the probabilistic perspective, presenting both theoretical analysis and empirical evidence to demonstrate how this expansion enhances attack effectiveness without compromising perceptual quality.

## 3.3 CONCEPT-BASED ADVERSARIAL ATTACKS GENERATE HIGHER QUALITY ADVERSARIAL EXAMPLES

From the probabilistic perspective, generating adversarial examples amounts to sampling from the overlap between $p_{\text{vic}}$ and $p_{\text{dis}}$, since $p_{\text{adv}}$ is proportional to their product (Hinton, 2002). For the common case of attacking a single original image $x_{\text{ori}}$, this procedure is illustrated on the left side of Figure 1. Empirical research has shown that modern robust classifiers can produce high-quality

---

[1] Both Zhang et al. (2024b) and our work present a methodology applicable to any PGM (e.g., VAE, energy-based, or diffusion). Since diffusion models are the most powerful PGMs, we adopt them throughout this paper. We use ProbAttack to denote the diffusion-based implementation of Zhang et al. (2024b).

images of the target classes (Santurkar et al., 2019; Zhang et al., 2024b; Zhu et al., 2021), causing $p_{\text{vic}}$ to concentrate on the semantics of those classes. Consequently, as $x_{\text{ori}}$ does not depict the target class, the intersection between $p_{\text{vic}}$ and $p_{\text{dis}}$ is small. Since high-quality images rarely appear in low-density regions of the distribution, the resulting adversarial examples drawn from this limited intersection tend to be of lower quality.

We claim that our concept-based adversarial attacks reduce the distance between the $p_{\text{vic}}$ and the $p_{\text{dis}}$, thereby increasing their overlap. This broader overlap yields higher-quality adversarial examples and improves targeted attack success rates, as illustrated on the right side of Figure 1.

To justify this claim, we must address two key questions:

- Do concept-based adversarial attacks indeed decrease the distance between $p_{\text{vic}}$ and $p_{\text{dis}}$?
- Do they genuinely produce better adversarial examples?

The remainder of this paper focuses on answering these questions.

### 3.4 THE DISTANCE BETWEEN DISTRIBUTIONS: A THEORETICAL STUDY

In the white-box adversarial attack scenario, both the victim classifier and target label are provided, which means $p_{\text{vic}}$ remains fixed. Let $p_{\text{dis}}(\cdot \mid \mathcal{C}_{\text{ori}})$ be a Gibbs distribution of the form $p_{\text{dis}}(x \mid \mathcal{C}_{\text{ori}}) \propto \exp(-\beta D(x, \mu))$, where $D$ is a distance function measuring the discrepancy between a point $x$ and the concept center $\mu$. The following theorem shows that, under suitable conditions, increasing the dispersion of $p_{\text{dis}}$ (i.e., decreasing $\beta^2$) reduces the KL divergence between $p_{\text{vic}}$ and $p_{\text{dis}}$:

**Theorem 1.** *Let $p$ be a probability distribution and $q$ be a Gibbs distribution of the form*

$$q(x) = \frac{\exp(-\beta D(x, \mu))}{Z(\beta)},$$

*where $Z(\beta)$ is the normalizing constant, $\mu$ is a constant and $D$ is a distance function. Then $KL(p \parallel q)$ is an increasing function of $\beta$ whenever $\mathbb{E}_{X \sim p}[D(X, \mu)] > \mathbb{E}_{X \sim q}[D(X, \mu)]$.*

The proof is provided in Appendix A. By Theorem 1, we see that $KL(p_{\text{vic}} \parallel p_{\text{dis}})$ decreases as $\beta$ decreases, provided that $\mathbb{E}_{X \sim p}[D(X, \mu)] > \mathbb{E}_{X \sim q}[D(X, \mu)]$. In the probabilistic adversarial attack framework, this condition is always satisfied because samples drawn from $p_{\text{vic}}$ lie farther from the mean of $p_{\text{dis}}$ than samples drawn from $p_{\text{dis}}$ itself. If this were not the case, the fundamental assumption that $p_{\text{dis}}$ represents a distance distribution concentrated around $x_{\text{ori}}$ or $\mathcal{C}_{\text{ori}}$ would be violated.

In practice, different PGMs may be used to model $p_{\text{dis}}$. When an energy-based model (EBM) is adopted, it explicitly learns a Gibbs distribution (LeCun et al., 2006). When a diffusion model (via score matching) is used, it instead learns an implicit representation of the corresponding energy function (Song & Ermon, 2019). Consequently, treating $p_{\text{dis}}$ as a Gibbs distribution in this section is fully consistent with practical implementations, and is also aligned with the probabilistic adversarial attack formulation in Section 2.3.

### 3.5 THE DISTANCE BETWEEN DISTRIBUTIONS: AN EMPIRICAL STUDY

The following theorem provides a tractable expression for the difference in KL divergence between a fixed victim distribution and two different distance distributions.

**Theorem 2.** *Let $p_{dis}^{(1)} = p_{dis}(\cdot \mid \mathcal{C}_{ori}^{(1)})$ and $p_{dis}^{(2)} = p_{dis}(\cdot \mid \mathcal{C}_{ori}^{(2)})$ be two distance distributions, and let $p_{vic}(\cdot \mid y_{tar})$ be the victim distribution corresponding to a victim classifier $p(y_{tar} \mid x)$. Then, the difference*

$$\Delta := KL\big(p_{dis}^{(1)} \parallel p_{vic}\big) \; - \; KL\big(p_{dis}^{(2)} \parallel p_{vic}\big)$$

*is given by*

$$\Delta = \mathbb{E}_{X \sim p_{dis}^{(1)}}\big[\log p_{dis}^{(1)}(X) - c \log p(y_{tar} \mid X)\big] - \mathbb{E}_{X \sim p_{dis}^{(2)}}\big[\log p_{dis}^{(2)}(X) - c \log p(y_{tar} \mid X)\big].$$

---

[2]A smaller $\beta$ corresponds to a higher "temperature" in the Gibbs distribution, which makes $p_{\text{dis}}$ more dispersed.

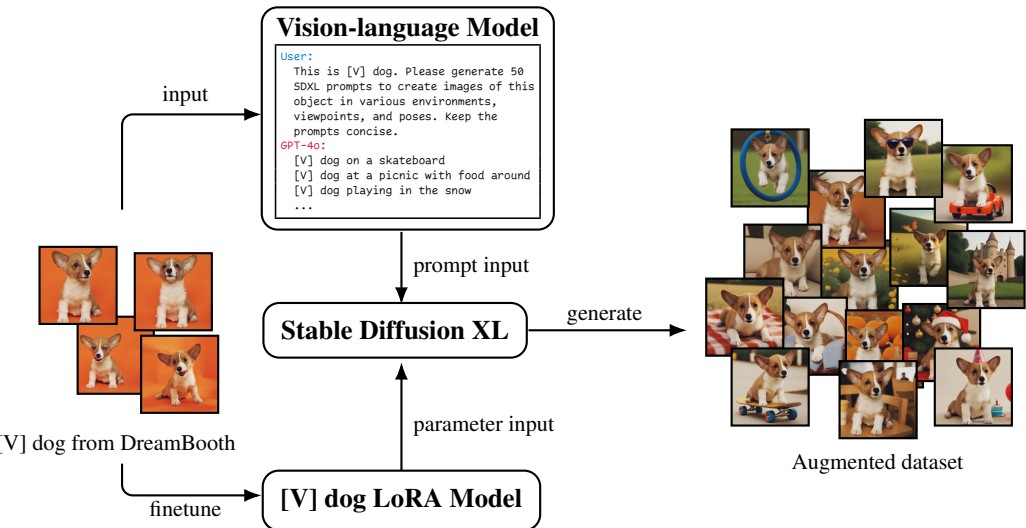

Figure 2: Illustration of how a single corgi concept ("[V] dog") is expanded into a diverse dataset. DreamBooth images (left) are finetuned with LoRA in Stable Diffusion XL, guided by GPT-4o prompts, to generate various poses, viewpoints, and environments (right).

The proof is provided in Appendix A. We estimate $\Delta$ via Monte Carlo integration and further reduce variance by using common random numbers in the sampling process; details of such practical techniques are introduced in Appendix B. In Section 5.3, we empirically show that, concept-based adversarial attacks reduce the distance between distributions $p_{\text{vic}}$ and $p_{\text{dis}}$ by showing $\Delta < 0$ when $p_{\text{dis}}^{(2)}$ is a distance distribution around only one image and $p_{\text{dis}}^{(1)}$ is a distance distribution around a concept.

# 4 GENERATING CONCEPT-BASED ADVERSARIAL EXAMPLES

In this section we introduce some practical methods to generate concept-based adversarial examples.

## 4.1 AUGMENT CONCEPT DATASETS BY MODERN GENERATIVE MODELS

In practice, it can be somewhat challenging to obtain a high-quality, highly diverse dataset $\mathcal{C}_{\text{ori}}$ depicting the same concept, as required by our method. For example, as shown on the left side of Figure 2, the dataset provided by DreamBooth (Ruiz et al., 2023) contains four images of the same long-eared corgi. Although the corgi is shown in various poses and from multiple viewpoints, the relatively uniform backgrounds do not provide sufficient diversity for our concept-based adversarial attack. Therefore, we decided to use Stable Diffusion XL (Podell et al., 2023) to expand the concept-description dataset.

As illustrated in Figure 2, we designate the corgi as "[V] dog". Using LoRA finetuning (Hu et al., 2022), we train an SDXL LoRA model on this concept. Next, we feed the five corgi images into GPT-4o (Hurst et al., 2024), stating that these images represent the "[V] dog" and asking it to produce SDXL prompts that embody sufficient diversity for the "[V] dog." Finally, we load the corgi LoRA into SDXL and, guided by GPT-4o's prompts (on the top of Figure 2), generate images featuring a wide range of viewpoints, environments, and poses for this corgi concept (see the right side of Figure 2).

## 4.2 Sample Selection

A key advantage of probabilistic adversarial attacks is that we can draw multiple samples from $p_{\text{adv}}$ and select the best ones[3]. In a white-box scenario, we can simply discard samples that fail to deceive the classifier (rejection sampling). However, if $p_{\text{dis}}$ and $p_{\text{vic}}$ overlap only slightly, this may lead to high rejection rates (especially under a top-1 success criterion).

As a workaround, we first sample $M$ adversarial examples from $p_{\text{adv}}$ and select the best among them. For small batches, it is feasible to manually choose which examples preserve the original concept. However, because we need a large number of adversarial examples, we use an automated approach: we sort the samples by how highly they rank the target class and, in the event of a tie, we employ one of two strategies — referred to here as the "conservative strategy" (CONS) and the "aggressive strategy" (AGGR). Under the conservative strategy, we pick the example with the lowest softmax probability, thereby filtering out samples that deviate significantly from the original concept. Under the aggressive strategy, we pick the example with the highest softmax probability, helping us select samples with the greatest adversarial potential.

## 5 Experiments

### 5.1 Data Preparation

We use the DreamBooth dataset (Ruiz et al., 2023), which provides 30 objects (animals, dolls, and everyday items) each with 5-6 representative images. To increase diversity, we apply the augmentation method in Section 4.1, generating 30 additional images per concept and forming the DreamBoothPlus dataset. Among the 30 objects of DreamBooth, we only augment 26, excluding four that pose challenges for text generation or require different fine-tuning parameters for cartoon-style content.

### 5.2 Fitting the Distance Distributions

We use the DreamBoothPlus dataset to finetune a diffusion model (Dhariwal & Nichol, 2021; Nichol & Dhariwal, 2021) to fit the distance distribution $p_{\text{dis}}$ (Details in Appendix D). We choose this model over more advanced architectures, such as the Stable Diffusion series (Podell et al., 2023; Rombach et al., 2022) or Flux, because it directly models $p(x)$ instead of $p(x \mid y)$, where $x$ is the image and $y$ is a label or prompt. Our goal is to employ a more principled model to illustrate our general adversarial attack method, rather than to optimize for the highest possible engineering performance.

### 5.3 Calculating the Difference between KL Divergences

We empirically estimate the difference between two KL divergences,

$$\Delta := KL(p_{\text{dis}}^{(1)} \, \| \, p_{\text{vic}}) \; - \; KL(p_{\text{dis}}^{(2)} \, \| \, p_{\text{vic}}),$$

by using the Monte-Carlo method and the practical techniques introduced in Section 3.5 and Appendix B, and we denote this estimate by $\tilde{\Delta}$. Concretely, for each concept in DreamBoothPlus, we fine-tune a diffusion model on the entire concept to obtain $p_{\text{dis}}^{(1)}$. We then fine-tune a separate diffusion model on just one image to obtain $p_{\text{dis}}^{(2)}$. Next, we calculate the empirical difference $\tilde{\Delta}$ and find that $\tilde{\Delta} < 0$ for every concept. This strongly suggests that $\Delta < 0$, confirming our hypothesis from Section 3.5. The table in Appendix B summarizes the values of $\tilde{\Delta}$ for each concept.

### 5.4 Generating Targeted Adversarial Examples

We evaluate the performance of the concept-based adversarial attack in a targeted adversarial attack setting, because targeted attacks are generally more difficult than untargeted attacks[4]. In our

---

[3]Although deterministic methods may yield different results when sampled multiple times, their variability does not stem from algorithmic design but rather from other sources of error.

[4]This is especially true for ImageNet classifiers, which must distinguish among 1,000 classes. In an untargeted attack, the goal is simply to prevent the victim classifier from assigning the adversarial sample to its

experiments, we compare NCF (Yuan et al., 2022), ACA (Chen et al., 2024b), DiffAttack (Chen et al., 2024a), and ProbAttack (Zhang et al., 2024b). We include NCF because it is the strongest color-based adversarial attack. Both ACA and DiffAttack apply adversarial gradients in the latent space induced by Stable Diffusion and DDIM, representing the state of the art in unrestricted adversarial attacks. As discussed earlier in Section 3, ProbAttack can be viewed as a special case of our approach when $|\mathcal{C}_{\text{ori}}| = 1$. For both ProbAttack and our approach, we set the number of samples $M$ to 10. During the sample selection phase of the experiments, our method uses two strategies — a conservative strategy and an aggressive strategy (both described in Section 4.2) — denoted in the tables as OURS (CONS) and OURS (AGGR), respectively.

For each compared method, we conduct a white-box attack on the victim classifier (also referred to as the surrogate classifier) by generating adversarial samples based on it. Next, we feed these white-box-generated adversarial samples into other classifiers — a process known as a black-box attack. If these additional classifiers also classify the adversarial samples into the target class, the black-box attack is deemed successful, indicating transferability.

In our experiments, ResNet50 (He et al., 2016) is used as the victim classifier for white-box attacks. We measure transferability on VGG19 (Simonyan & Zisserman, 2015), ResNet152 (He et al., 2016), DenseNet161 (Huang et al., 2017), Inception V3 (Szegedy et al., 2016), EfficientNet B7 (Tan & Le, 2019), and on adversarially trained Inception V3 Adv (Kurakin et al., 2017), EfficientNet B7 Adv (Xie et al., 2020), and Ensemble IncRes V2 (Tramèr et al., 2018). We report both the white-box targeted attack success rate (on ResNet50) and the black-box transfer success rate (on the remaining models).

Table 1: Targeted attack success rates (%) on ImageNet classifiers. In the white-box setting, a targeted attack is counted as successful if the target class is ranked first. For transferability, we report top-5 success rates, counting an attack as successful if the target class is among the top 5 predictions (since top-1 success was uniformly low across all methods). See Appendix Q for full results.

| | NCF | ACA | DiffAttack | ProbAttack | OURS (CONS) | OURS (AGGR) |
|---|---|---|---|---|---|---|
| **White-box** | | | **Targeted-Top1** | | | |
| ResNet 50 | 1.15 | 6.03 | 84.23 | 59.23 | **97.82** | **97.82** |
| **Transferability** | | | **Targeted-Top5** | | | |
| VGG19 | 1.28 | 1.67 | **4.36** | 2.44 | 2.05 | **4.36** |
| ResNet 152 | 1.41 | 1.92 | 8.33 | 3.33 | 2.82 | **8.72** |
| DenseNet 161 | 1.41 | 2.05 | 7.44 | 3.97 | 3.85 | **11.54** |
| Inception V3 | 0.90 | 1.41 | 3.08 | 2.56 | 1.28 | **4.74** |
| EfficientNet B7 | 1.41 | 1.67 | 1.79 | 1.41 | 1.28 | **3.97** |
| **Adversarial Defence** | | | | | | |
| Inception V3 Adv | 1.15 | 1.28 | 3.21 | 2.18 | 0.90 | **3.72** |
| EfficientNet B7 Adv | 0.26 | 1.15 | 2.05 | 2.31 | 1.67 | **6.41** |
| Ensemble IncRes V2 | 0.77 | 1.28 | 2.69 | 1.92 | 0.77 | **5.00** |

For unrestricted adversarial examples, we must also check whether they preserve the original concept and remain undetectable to humans. Therefore, we measure similarity via a user study (Appendix F) and CLIP (Radford et al., 2021), and image quality using no-reference metrics (MUSIQ (Ke et al., 2021), TReS (Golestaneh et al., 2022), NIMA (Talebi & Milanfar, 2018), ARNIQA (Agnolucci et al., 2024), DBCNN (Zhang et al., 2020), and HyperIQA (Su et al., 2020)).

As ImageNet has 1,000 classes, it is impractical to evaluate them all. Therefore, we randomly select 30 target classes $y_{\text{tar}}$, listed in Appendix E. Since DreamBoothPlus contains 26 concepts, each method generates $26 \times 30 = 780$ adversarial examples. This scale is comparable to current popular approaches performing untargeted adversarial attacks on the ImageNet-Compatible dataset (Kurakin et al., 2018).

Table 1 presents the targeted attack success rates on ImageNet classifiers. Since the choice of aggressive or conservative strategy in the sample selection phase does not affect white-box performance,

---

correct class. In contrast, a targeted attack requires the classifier to misclassify the adversarial sample exactly as the chosen target class $y_{\text{tar}}$.

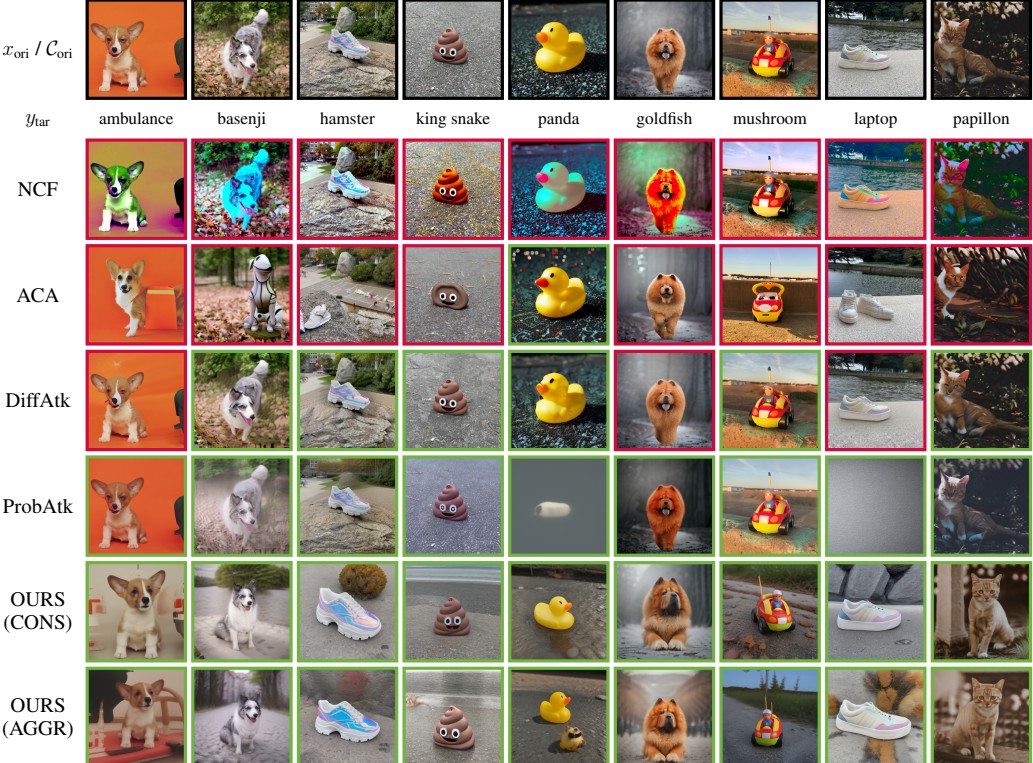

Figure 3: Qualitative comparison. (A **green** border indicates an example that successfully fools the classifier; **red** indicates failure.) See Appendix G for a more detailed qualitative analysis.

those rates are identical. Notably, the aggressive strategy achieves significantly higher transferability than other methods. While the conservative strategy leads to slightly lower transferability, it is still roughly comparable to the baseline methods. Please refer to Appendix Q for full results.

Table 2 reports the similarity between each adversarial sample and its original image, as well as the image quality of the generated examples. Both our aggressive and conservative strategies outperform the other methods in these metrics. Combined with the attack success rates in Table 1, our approach not only achieves higher success but also preserves the original concept $\mathcal{C}_{ori}$ more effectively. Figure 3 provides a qualitative comparison, showing how well our method maintains the original concept. Notably, DiffAttack generates images missing details, which aligns with its weaker image quality scores. For additional qualitative analysis, please refer to Appendix G.

Table 2: Quantitative comparison of similarity to the original images and no reference image quality metrics for unrestricted adversarial examples.

|  | Clean | NCF | ACA | DiffAttack | ProbAttack | OURS (CONS) | OURS (AGGR) |
|---|---|---|---|---|---|---|---|
| **Similarity** |  |  |  |  |  |  |  |
| ↑ User Study | N/A | 0.1859 | 0.2808 | 0.7577 | 0.8041 | **0.9654** | 0.8808 |
| ↑ Avg. Clip Score | 1.0 | **0.8728** | 0.7861 | 0.8093 | 0.8581 | 0.8283 | 0.8043 |
| **Image Quality** |  |  |  |  |  |  |  |
| ↑ HyperIQA | 0.7255 | 0.5075 | 0.6462 | 0.5551 | 0.6675 | **0.6947** | 0.6809 |
| ↑ DBCNN | 0.6956 | 0.5096 | 0.6103 | 0.5294 | 0.6161 | **0.6572** | 0.6399 |
| ↑ ARNIQA | 0.7667 | 0.5978 | 0.6879 | 0.6909 | 0.7009 | **0.7335** | 0.7154 |
| ↑ MUSIQ-AVA | 4.3760 | 3.8135 | 4.2687 | 4.0734 | 4.3130 | **4.5305** | 4.5250 |
| ↑ NIMA-AVA | 4.5595 | 3.7916 | 4.4511 | 4.0589 | 4.5168 | **4.7575** | 4.7401 |
| ↑ MUSIQ-KonIQ | 65.0549 | 50.5022 | 59.0840 | 52.5399 | 58.1563 | **63.7486** | 62.2217 |
| ↑ TReS | 93.2127 | 64.7050 | 85.8435 | 74.1167 | 84.3131 | **90.4488** | 88.0836 |

## 6 RELATED WORKS

To the best of our knowledge, no existing method constructs adversarial examples conditioned on an identity-level concept. The most closely related works are Song et al. (2018)'s, Dai et al. (2024)'s and Collins et al. (2025)'s. However, they treat a "class" (e.g., cat, dog, or truck) as the concept, which cannot precisely capture an individual identity. In our framework, we represent a concept through a distribution that could be learned from a set of images, allowing it to flexibly correspond to a single image (i.e. a set with size 1), an identity-level concept, or a class-level concept. By contrast, other unrestricted adversarial attack methods focus solely on generating adversarial examples from a single image: Xiao et al. (2018a) generate adversarial examples using GANs, Chen et al. (2023) employ diffusion models, and Laidlaw et al. (2020) impose a feature-space distance as a regularization term in the optimization objective. ACA (Chen et al., 2024b) and DiffAttack (Chen et al., 2024a) further apply the attack gradient directly in the DDIM latent space (Song et al., 2020). Color-based transformations have proven effective in preserving semantic content for untargeted attacks (Bhattad et al., 2020; Hosseini & Poovendran, 2018; Shamsabadi et al., 2020; Yuan et al., 2022; Zhao et al., 2020), yet they perform poorly in targeted scenarios (Chen et al., 2024a), a result confirmed by our experiments.

| Unrestricted Adversarial Attack Method | Single-image | Identity-level Concept | Class-level Concept |
|---|---|---|---|
| Bhattad et al. (2020) | Yes | No | No |
| Hosseini & Poovendran (2018) | Yes | No | No |
| Colorfool (Shamsabadi et al., 2020) | Yes | No | No |
| Zhao et al. (2020) | Yes | No | No |
| NCF (Yuan et al., 2022) | Yes | No | No |
| ACA (Chen et al., 2024b) | Yes | No | No |
| DiffAttack (Chen et al., 2024a) | Yes | No | No |
| ProbAttack (Zhang et al., 2024b) | Yes | No | No |
| AdvGAN (Xiao et al., 2018a) | Yes | No | No |
| Xiao et al. (2018b) | Yes | No | No |
| Perceptual Adv. Attack (Laidlaw et al., 2020) | Yes | No | No |
| Song et al. (2018) | No | No | Yes |
| NatADiff (Collins et al., 2025) | No | No | Yes |
| AdvDiff (Dai et al., 2024) | No | No | Yes |
| AdvDiffuser Chen et al. (2023) | Yes | No | Yes |
| **Ours: Concept-based Adv. Attack** | Yes | Yes | Yes |

Table 3: Comparison of unrestricted adversarial attack methods. Our method is the only one capable of performing adversarial attacks at the identity-level concept, while also supporting single-image and class-level concepts.

Our work directly inherited from Zhang et al. (2024b)'s probabilistic perspective, but we make a novel contribution by, for the first time, defining the distance distribution in adversarial attacks with respect to a distribution representing a concept rather than a single image. Although, operationally, our method appears to be a straightforward extension — replacing the single-image-centered distribution $p_{\text{dis}}(\cdot \mid x_{\text{ori}})$ with a concept-centered distribution $p_{\text{dis}}(\cdot \mid C_{\text{ori}})$ (as in the difference between (2) and (3)) — we rigorously demonstrate, both theoretically and empirically, why this seemingly simple generalization is remarkably effective.

## 7 CONCLUSIONS

The essence of adversarial attacks is to create examples that are imperceptible to humans yet harmful to computational systems. Our work demonstrates that in an era of powerful generative models, creating an adversarial example from scratch — one that humans perceive as conceptually correct — can be more flexible, more realistic, and ultimately more potent than simply perturbing a single image. Leveraging modern generative models, adversarial noise can be concealed in subtle changes to viewpoint, pose, or background, making it exceedingly difficult to detect. We believe that our concept-based adversarial attack heralds the future of adversarial attacks, posing new challenges to the field of AI security. Defending against such threats will be crucial for advancing AI security research.

## ACKNOWLEDGMENTS

This work was supported by the Engineering and Physical Sciences Research Council (EPSRC) through the AI Hub in Generative Models (EP/Y028805/1).

AZ and SK were funded by the UKRI AI Hub in Generative Models (EP/Y028805/1). SM was supported by the UKRI AI Programme Grant (EP/Y028856/1). AZ also acknowledges support from a personal grant provided by Mrs. Yanshu Wu. SK was additionally supported by the Research Council of Finland (Flagship Programme: Finnish Center for Artificial Intelligence FCAI, 359207) and the UKRI Turing AI World-Leading Researcher Fellowship (EP/W002973/1).

## ETHICS STATEMENT

This work introduces a new class of adversarial attacks that operate at the concept level. While our primary goal is to advance scientific understanding of adversarial robustness and stimulate the development of stronger defenses, we acknowledge the potential for malicious misuse. In particular, concept-based adversarial attacks could be exploited to evade security-sensitive image classifiers or to manipulate systems deployed in safety-critical applications.

To mitigate these risks, we have:

- Released all code and data strictly for research purposes, under licenses that encourage responsible use.

- Discussed mitigation strategies in Appendix L, including adversarial training, AI-generated content detection, and hybrid defenses.

We emphasize that the broader impact (Appendix M) of this work depends on the research community's response. By exposing vulnerabilities of current classifiers, we aim to encourage the development of more robust and trustworthy AI systems. We strongly discourage any use of this research for harmful purposes.

## REPRODUCIBILITY STATEMENT

We have taken several steps to ensure the reproducibility of our work:

- **Code and models**: We provide the full source code, including scripts for dataset preparation, model fine-tuning, and adversarial example generation, at `https://github.com/andiac/ConceptAdv`. All hyperparameters and training details are specified in the code repository.

- **Datasets**: Our experiments are based on the DreamBooth dataset (Ruiz et al., 2023), which is publicly available under a CC-BY-4.0 license. We also describe our augmentation procedure using SDXL and LoRA in Section 4.1, and provide scripts for generating data (DreamBoothPlus) as part of our code repo.

- **Hyperparameters**: Fine-tuning settings for both SDXL LoRA and diffusion models are detailed in Appendix D. For sampling and evaluation, we report the number of generated adversarial examples, sampling strategies, and evaluation metrics in Sections 4-5.

- **Theoretical results**: Proofs of all theorems are included in Appendix A, and additional details on KL divergence estimation are in Appendix B.

- **Compute resources**: We report hardware specifications and training times in Appendix I to allow others to reproduce our experiments with similar resources.

We believe these resources provide sufficient detail for reproducing both our theoretical and empirical results.

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

# APPENDIX

## A   PROOF OF THE THEOREMS

**Theorem 1.** *Let $p$ be a probability distribution and $q$ be a Gibbs distribution of the form*

$$q(x) = \frac{\exp(-\beta D(x, \mu))}{Z(\beta)},$$

*where $Z(\beta)$ is the normalizing constant, $\mu$ is a constant and $D$ is a distance function. Then $KL(p \,\|\, q)$ is an increasing function of $\beta$ whenever $\mathbb{E}_{X \sim p}[D(X, \mu)] > \mathbb{E}_{X \sim q}[D(X, \mu)]$.*

*Proof.* According to the definition of KL divergence, we have

$$KL(p\|q) = \int p(x) \log \frac{p(x)}{q(x)} dx = \int p(x) \log p(x) dx - \int p(x) \log q(x) dx$$

Since $\int p(x) \log p(x) dx$ is independent of $\beta$, we can treat it as a constant. Let us denote the $\beta$-dependent component as $f(\beta)$, which gives us

$$
\begin{aligned}
f(\beta) &= - \int p(x) \log q(x) dx \\
&= - \int p(x) \left[ -\beta D(x, \mu) - \log Z(\beta) \right] dx \\
&= \int p(x) \beta D(x, \mu) dx + \log Z(\beta) \int p(x) dx \\
&= \mathbb{E}_{X \sim p}[\beta D(X, \mu)] + \log Z(\beta)
\end{aligned}
$$

Taking the derivative with respect to $\beta$, we obtain

$$
\begin{aligned}
\frac{d}{d\beta} f(\beta) &= \mathbb{E}_{X \sim p}[D(X, \mu)] + \frac{1}{Z(\beta)} \frac{dZ(\beta)}{d\beta} \\
&= \mathbb{E}_{X \sim p}[D(X, \mu)] + \frac{1}{Z(\beta)} \int \frac{d \exp(-\beta D(X, \mu))}{d\beta} dx \\
&= \mathbb{E}_{X \sim p}[D(X, \mu)] + \frac{1}{Z(\beta)} \int -D(X, \mu) \exp(-\beta D(X, \mu)) dx \\
&= \mathbb{E}_{X \sim p}[D(X, \mu)] - \mathbb{E}_{X \sim q}[D(X, \mu)]
\end{aligned}
$$

Therefore, when $\mathbb{E}_{X \sim p}[D(X, \mu)] > \mathbb{E}_{X \sim q}[D(X, \mu)]$, the derivative becomes positive. This implies that both $f(\beta)$ and consequently $KL(p\|q)$ increase as $\beta$ increases. □

**Theorem 2.** *Let $p_{dis}^{(1)} = p_{dis}(\cdot \mid \mathcal{C}_{ori}^{(1)})$ and $p_{dis}^{(2)} = p_{dis}(\cdot \mid \mathcal{C}_{ori}^{(2)})$ be two distance distributions, and let $p_{vic}(\cdot \mid y_{tar})$ be the victim distribution corresponding to a victim classifier $p(y_{tar} \mid x)$. Then, the difference*

$$\Delta = KL\big(p_{dis}^{(1)} \,\|\, p_{vic}\big) \; - \; KL\big(p_{dis}^{(2)} \,\|\, p_{vic}\big)$$

*is given by*

$$\Delta = \mathbb{E}_{X \sim p_{dis}^{(1)}} \big[ \log p_{dis}^{(1)}(X) - c \log p(y_{tar} \mid X) \big] - \mathbb{E}_{X \sim p_{dis}^{(2)}} \big[ \log p_{dis}^{(2)}(X) - c \log p(y_{tar} \mid X) \big].$$

*Proof.* Recall that for distributions $p$ and $q$ on the same space, the Kullback–Leibler (KL) divergence is

$$KL(p \,\|\, q) \; = \; \mathbb{E}_{X \sim p} \big[ \log p(X) \; - \; \log q(X) \big].$$

Hence, for $p_{dis}(\cdot \mid \mathcal{C}_{ori})$ and $p_{vic}(\cdot \mid y_{tar})$,

$$KL\big(p_{dis}(\cdot \mid \mathcal{C}_{ori}) \,\|\, p_{vic}(\cdot \mid y_{tar})\big) \; = \; \mathbb{E}_{X \sim p_{dis}(\cdot \mid \mathcal{C}_{ori})} \big[ \log p_{dis}(X \mid \mathcal{C}_{ori}) \; - \; \log p_{vic}(X \mid y_{tar}) \big].$$

Given that the victim distribution is proportional to

$$p_{\text{vic}}(x \mid y_{\text{tar}}) \; \propto \; \exp\!\big(-c\,f(x, y_{\text{tar}})\big),$$

where $f(x, y_{\text{tar}})$ is the cross-entropy loss, i.e. $f(x, y_{\text{tar}}) = -\log p(y_{\text{tar}} \mid x)$. Thus we can write

$$p_{\text{vic}}(x \mid y_{\text{tar}}) \; = \; \frac{\exp(-c\,f(x, y_{\text{tar}}))}{\int \exp(-c\,f(x, y_{\text{tar}}))\,dx}.$$

Let $Z := \int \exp\!\big(-c\,f(x, y_{\text{tar}})\big)\,dx$. Then

$$\log p_{\text{vic}}(X \mid y_{\text{tar}}) \; = \; \log \exp\!\big(-c\,f(X, y_{\text{tar}})\big) \; - \; \log Z \; = \; -c\,f(X, y_{\text{tar}}) \; - \; \log Z.$$

Therefore,

$$KL\big(p_{\text{dis}}(\cdot \mid \mathcal{C}_{\text{ori}}) \,\|\, p_{\text{vic}}(\cdot \mid y_{\text{tar}})\big) \; = \; \mathbb{E}_{X \sim p_{\text{dis}}(\cdot|\mathcal{C}_{\text{ori}})}\big[\log p_{\text{dis}}(X \mid \mathcal{C}_{\text{ori}}) \; + \; c\,f(X, y_{\text{tar}})\big] \; + \; \log Z.$$

Since $f(x, y_{\text{tar}}) = -\log p(y_{\text{tar}} \mid x)$, we get

$$c\,\mathbb{E}_{X \sim p_{\text{dis}}(\cdot|\mathcal{C}_{\text{ori}})}[f(X, y_{\text{tar}})] \; = \; -c\,\mathbb{E}_{X \sim p_{\text{dis}}(\cdot|\mathcal{C}_{\text{ori}})}[\log p(y_{\text{tar}} \mid X)].$$

Hence

$$KL\big(p_{\text{dis}}(\cdot \mid \mathcal{C}_{\text{ori}}) \,\|\, p_{\text{vic}}(\cdot \mid y_{\text{tar}})\big) \; = \; \mathbb{E}_{X \sim p_{\text{dis}}(\cdot|\mathcal{C}_{\text{ori}})}\big[\log p_{\text{dis}}(X \mid \mathcal{C}_{\text{ori}}) \; - \; c\,\log p(y_{\text{tar}} \mid X)\big] \; + \; \log Z.$$

Now take two such distributions, $p_{\text{dis}}^{(1)}$ and $p_{\text{dis}}^{(2)}$. Because $\log Z$ does not depend on which $p_{\text{dis}}(\cdot \mid \mathcal{C}_{\text{ori}})$ we use, it cancels when we form the difference:

$$
\begin{aligned}
\Delta = {} & KL\big(p_{\text{dis}}^{(1)} \,\|\, p_{\text{vic}}\big) \; - \; KL\big(p_{\text{dis}}^{(2)} \,\|\, p_{\text{vic}}\big) \\
= {} & \Big(\mathbb{E}_{X \sim p_{\text{dis}}^{(1)}}\big[\log p_{\text{dis}}^{(1)}(X) - c\log p(y_{\text{tar}} \mid X)\big] + \log Z\Big) \\
& - \Big(\mathbb{E}_{X \sim p_{\text{dis}}^{(2)}}\big[\log p_{\text{dis}}^{(2)}(X) - c\log p(y_{\text{tar}} \mid X)\big] + \log Z\Big) \\
= {} & \mathbb{E}_{X \sim p_{\text{dis}}^{(1)}}\big[\log p_{\text{dis}}^{(1)}(X) \; - \; c\,\log p(y_{\text{tar}} \mid X)\big] \; - \; \mathbb{E}_{X \sim p_{\text{dis}}^{(2)}}\big[\log p_{\text{dis}}^{(2)}(X) \; - \; c\,\log p(y_{\text{tar}} \mid X)\big],
\end{aligned}
$$

which is precisely the claimed result. $\qquad\square$

## B   PRACTICAL STRATEGIES FOR KL DIVERGENCE ESTIMATION

### B.1   COMMON RANDOM NUMBERS

In practice, our distance distributions $p_{\text{dis}}^{(1)}$ and $p_{\text{dis}}^{(2)}$ are instantiated by diffusion models: each image $X$ is generated from noise $\epsilon$ via a generator $\mathcal{G}$. We write $X = \mathcal{G}^{(1)}(\epsilon)$ to indicate that $X$ is sampled from $p_{\text{dis}}^{(1)}$, and $X = \mathcal{G}^{(2)}(\epsilon)$ to indicate that $X$ is sampled from $p_{\text{dis}}^{(2)}$. In this common-noise setup, the difference in KL divergences becomes

$$\Delta = \mathbb{E}_{\epsilon}\Big[\log p_{\text{dis}}^{(1)}(\mathcal{G}^{(1)}(\epsilon)) - c\log p(y_{\text{tar}} \mid \mathcal{G}^{(1)}(\epsilon)) - \log p_{\text{dis}}^{(2)}(\mathcal{G}^{(2)}(\epsilon)) + c\log p(y_{\text{tar}} \mid \mathcal{G}^{(2)}(\epsilon))\Big]. \quad (4)$$

### B.2   LIKELIHOOD CORRECTION

We posit a probabilistic model $p_{\text{share}}$ that captures the non-semantic, shared features of the images. Specifically, we assume that for samples $X$ drawn from $p_{\text{dis}}^{(1)}$ and $p_{\text{dis}}^{(2)}$, the expected values $\mathbb{E}_{X \sim p_{\text{dis}}^{(1)}}[\log p_{\text{share}}(X)]$ and $\mathbb{E}_{X \sim p_{\text{dis}}^{(2)}}[\log p_{\text{share}}(X)]$ are equal. This assumption is reasonable because the two distributions, in principle, can generate images sharing the same non-semantic details, differing only in their semantic content.

Then, under our diffusion-model instantiation, we have

$$\mathbb{E}_{X \sim p_{\text{dis}}^{(1)}}[\log p_{\text{share}}(X)] = \mathbb{E}_{X \sim p_{\text{dis}}^{(2)}}[\log p_{\text{share}}(X)] \Rightarrow \mathbb{E}_{\epsilon}\left[\log p_{\text{share}}(\mathcal{G}^{(1)}(\epsilon)) - \log p_{\text{share}}(\mathcal{G}^{(2)}(\epsilon))\right] = 0$$

for $\Delta$, following equation 4, we have the equations:

$$\Delta = \mathbb{E}_\epsilon \left[ \log p_{\mathrm{dis}}^{(1)}(\mathcal{G}^{(1)}(\epsilon)) - c \log p(y_{\mathrm{tar}} \mid \mathcal{G}^{(1)}(\epsilon)) - \log p_{\mathrm{dis}}^{(2)}(\mathcal{G}^{(2)}(\epsilon)) + c \log p(y_{\mathrm{tar}} \mid \mathcal{G}^{(2)}(\epsilon)) \right]$$
$$- \mathbb{E}_\epsilon \left[ \log p_{\mathrm{share}}(\mathcal{G}^{(1)}(\epsilon)) - \log p_{\mathrm{share}}(\mathcal{G}^{(2)}(\epsilon)) \right]$$
$$= \mathbb{E}_\epsilon \left[ \log p_{\mathrm{dis}}^{(1)}(\mathcal{G}^{(1)}(\epsilon)) - \log p_{\mathrm{share}}(\mathcal{G}^{(1)}(\epsilon)) - c \log p(y_{\mathrm{tar}} \mid \mathcal{G}^{(1)}(\epsilon)) \right.$$
$$\left. - \log p_{\mathrm{dis}}^{(2)}(\mathcal{G}^{(2)}(\epsilon)) + \log p_{\mathrm{share}}(\mathcal{G}^{(2)}(\epsilon)) + c \log p(y_{\mathrm{tar}} \mid \mathcal{G}^{(2)}(\epsilon)) \right]$$

In practice, however, the assumption $\mathbb{E}_{X \sim p_{\mathrm{dis}}^{(1)}}[p_{\mathrm{share}}(X)] = \mathbb{E}_{X \sim p_{\mathrm{dis}}^{(2)}}[p_{\mathrm{share}}(X)]$ may not hold perfectly, due to differences in model capacity, finetuning steps, and other factors. For example, a model $p_{\mathrm{dis}}^{(1)}$ finetuned on more images might actually lose some non-semantic details compared to $p_{\mathrm{dis}}^{(2)}$, which is finetuned on a single image. The extra $p_{\mathrm{share}}$-based terms can thus be viewed as a correction that accounts for these mismatches in non-semantic content.

Following Zhang et al. (2024a)'s work, a good practical choice for $p_{\mathrm{share}}$ is often the pretrained model, because it has been trained on a large, diverse dataset and therefore captures broad, shared non-semantic features of images.

### B.3 ESTIMATED DIFFERENCES OF KL DIVERGENCES

Because $\Delta$ depends on both the original concept $\mathcal{C}_{\mathrm{ori}}$ and the target class $y_{\mathrm{tar}}$, enumerating every possible $\Delta$ would be impractical. Therefore, similar to the adversarial-example generation experiment introduced in Section 5.4, we restrict our analysis to the 30 target classes listed in Appendix E. We then compute the mean and variance of the estimated $\Delta$ (denoted $\tilde{\Delta}$), as shown in Table 4.

Table 4: Estimated differences of KL divergences $\tilde{\Delta}$ for each concept.

| Concept name | $\tilde{\Delta}$ | Concept name | $\tilde{\Delta}$ | Concept name | $\tilde{\Delta}$ |
|---|---|---|---|---|---|
| backpack | $-3507.28 \pm 2040.85$ | backpack_dog | $-8372.57 \pm 860.92$ | bear_plushie | $-466.69 \pm 1413.40$ |
| candle | $-3637.39 \pm 940.80$ | cat | $-4658.98 \pm 2423.05$ | cat2 | $-6654.28 \pm 1318.03$ |
| colorful_sneaker | $-4954.90 \pm 420.40$ | dog | $-7159.63 \pm 696.25$ | dog2 | $-4814.63 \pm 470.95$ |
| dog3 | $-8303.66 \pm 2964.40$ | dog5 | $-8080.70 \pm 2808.13$ | dog6 | $-2380.29 \pm 1780.36$ |
| dog7 | $-10545.45 \pm 787.79$ | dog8 | $-12401.76 \pm 966.62$ | duck_toy | $-2420.45 \pm 2334.07$ |
| fancy_boot | $-5780.99 \pm 1489.92$ | grey_sloth_plushie | $-6848.96 \pm 1477.16$ | monster_toy | $-7435.54 \pm 1698.99$ |
| pink_sunglasses | $-5711.26 \pm 799.22$ | poop_emoji | $-219.27 \pm 929.88$ | rc_car | $-5046.80 \pm 3212.23$ |
| robot_toy | $-7008.27 \pm 710.45$ | shiny_sneaker | $-3203.81 \pm 4447.21$ | teapot | $-7327.87 \pm 486.00$ |
| vase | $-8806.76 \pm 2435.28$ | wolf_plushie | $-325.80 \pm 2811.99$ | | |

## C SENSITIVITY STUDY FOR SAMPLE SELECTION

As described in Section 4.2, when generating adversarial examples we first sample $M$ candidate adversarial images and then select the "best" one. To investigate the effect of different values of $M$, we perform a sensitivity study. In this study, $|\mathcal{C}_{\mathrm{ori}}|$ is set to either 1 or 30, while $M$ is set to 1, 5, or 10. Note that $|\mathcal{C}_{\mathrm{ori}}| = 1$ corresponds to ProbAttack (Zhang et al., 2024b).

From Table 5, we observe that as $M$ increases, the white-box attack success rate rises significantly. In the $|\mathcal{C}_{\mathrm{ori}}| = 1$ case, the transferability also increases as $M$ grows. However, in the $|\mathcal{C}_{\mathrm{ori}}| = 30$ case, increasing $M$ results in a decrease in transferability. This happens because we employ a conservative strategy (introduced in Section 4.2) to pick images that just barely fool the classifier while preserving the original concept. Evidence of this conservative strategy can be seen in Table 6: when $|\mathcal{C}_{\mathrm{ori}}| = 30$, increasing $M$ yields higher image quality and greater similarity to the original image. In contrast, when $|\mathcal{C}_{\mathrm{ori}}| = 1$, the ranking criterion is more influential during the sample selection stage, so the conservative strategy does not take effect. Consequently, as $M$ grows, the image quality tends to decrease.

Table 5: Targeted attack success rates (%) on ImageNet classifiers. In the white-box setting, success is counted when the target class is the top prediction. For transferability, we report top 100 success rates, as top 1 success was uniformly low across all methods.

| | $|\mathcal{C}_{ori}| = 1$ $M = 1$ | $|\mathcal{C}_{ori}| = 1$ $M = 5$ | $|\mathcal{C}_{ori}| = 1$ $M = 10$ | $|\mathcal{C}_{ori}| = 30$ $M = 1$ | $|\mathcal{C}_{ori}| = 30$ $M = 5$ | $|\mathcal{C}_{ori}| = 30$ $M = 10$ |
|---|---|---|---|---|---|---|
| **White-box** | | | | **Top1** | | |
| Resnet 50 | 26.03 | 50.38 | 59.23 | 81.41 | 96.28 | **97.82** |
| **Transferability** | | | | **Top100** | | |
| VGG19 | 16.41 | 17.82 | 18.97 | **30.00** | 23.21 | 20.90 |
| ResNet 152 | 21.79 | 23.97 | 26.79 | **43.85** | 37.69 | 35.13 |
| DenseNet 161 | 26.67 | 30.26 | 33.08 | **53.08** | 44.10 | 41.03 |
| Inception V3 | 16.03 | 17.44 | 18.21 | **22.95** | 19.36 | 19.87 |
| EfficientNet B7 | 16.15 | 16.54 | 17.44 | **25.90** | 21.54 | 20.38 |
| **Adversarial Defence** | | | | **Top100** | | |
| Inception V3 Adv | 18.97 | 17.82 | 20.00 | **25.26** | 20.64 | 20.51 |
| EfficientNet B7 Adv | 16.54 | 17.95 | 21.03 | **33.72** | 27.82 | 24.74 |
| Ensemble IncRes V2 | 14.10 | 14.87 | 17.31 | **24.74** | 18.08 | 17.44 |

Table 6: Quantitative comparison of similarity to the original images and no reference image quality metrics for unrestricted adversarial examples.

| | Clean | $|\mathcal{C}_{ori}| = 1$ $M = 1$ | $|\mathcal{C}_{ori}| = 1$ $M = 5$ | $|\mathcal{C}_{ori}| = 1$ $M = 10$ | $|\mathcal{C}_{ori}| = 30$ $M = 1$ | $|\mathcal{C}_{ori}| = 30$ $M = 5$ | $|\mathcal{C}_{ori}| = 30$ $M = 10$ |
|---|---|---|---|---|---|---|---|
| **Similarity** | | | | | | | |
| ↑ User Study | N/A | N/A | N/A | 0.8041 | N/A | N/A | **0.9654** |
| ↑ Avg. Clip Score | 1.0 | **0.8953** | 0.8859 | 0.8581 | 0.8175 | 0.8286 | 0.8283 |
| **Image Quality** | | | | | | | |
| ↑ MUSIQ-KonIQ | 65.0549 | 59.0779 | 59.7026 | 58.1563 | 62.8293 | **63.8795** | 63.7486 |
| ↑ MUSIQ-AVA | 4.3760 | 4.3016 | 4.3400 | 4.3130 | 4.5013 | **4.5356** | 4.5305 |
| ↑ TReS | 93.2127 | 86.4084 | 86.9480 | 84.3131 | 88.9667 | **90.6312** | 90.4488 |
| ↑ NIMA-AVA | 4.5595 | 4.5729 | 4.5851 | 4.5168 | 4.6804 | 4.7422 | **4.7575** |
| ↑ HyperIQA | 0.7255 | 0.6800 | 0.6808 | 0.6675 | 0.6880 | **0.6952** | 0.6947 |
| ↑ DBCNN | 0.6956 | 0.6303 | 0.6340 | 0.6161 | 0.6459 | 0.6564 | **0.6572** |
| ↑ ARNIQA | 0.7667 | 0.7222 | 0.7153 | 0.7009 | 0.7187 | 0.7323 | **0.7335** |

# D   FINETUNING DETAILS

In this section, we provide the key parameters required to fine-tune the models. For all parameters, please refer to the code repository of this paper.

## D.1   FINETUNING DETAILS OF SDXL LORAS

In the SDXL LoRA finetuning described in Section 4.1 and Section 5.1, we use a LoRA rank of 128 with a corresponding LoRA alpha of 128, and set the dropout rate to 0.05. We employ the AdamW optimizer and train for 250 epochs, using a learning rate of $10^{-4}$ for UNet parameters and $10^{-5}$ for text encoder parameters. For more detailed settings, please refer to the accompanying code.

## D.2   FINETUNING DETAILS OF DISTANCE DISTRIBUTIONS

For the diffusion model fine-tuning described in Section 5.2, we set the learning rate to $10^{-6}$, use the AdamW optimizer, and train for 8000 steps per image in the fine-tuning set. More details are in the code repository.

# E   LIST OF THE SELECTED TARGET CLASSES

Since ImageNet consists of 1000 classes and it is impractical to cover them all, we randomly selected 30 target classes. For details, please refer to Table 7.

Table 7: List of the selected target classes

| Index | Description |
|-------|-------------|
| 1 | goldfish, Carassius auratus |
| 2 | great white shark, white shark, man-eater, man-eating shark, Carcharodon carcharias |
| 7 | cock |
| 56 | king snake, kingsnake |
| 134 | crane |
| 151 | Chihuahua |
| 157 | papillon |
| 231 | collie |
| 254 | basenji |
| 309 | bee |
| 328 | sea urchin |
| 333 | hamster |
| 341 | hog, pig, grunter, squealer, Sus scrofa |
| 345 | hippopotamus, hippo, river horse, Hippopotamus amphibius |
| 368 | gibbon, Hylobates lar |
| 388 | giant panda, panda, panda bear, coon bear, Ailuropoda melanoleuca |
| 404 | airliner |
| 407 | ambulance |
| 417 | balloon |
| 504 | coffee mug |
| 555 | fire engine, fire truck |
| 563 | fountain pen |
| 620 | laptop, laptop computer |
| 721 | pillow |
| 769 | rule, ruler |
| 817 | sports car, sport car |
| 894 | wardrobe, closet, press |
| 947 | mushroom |
| 955 | jackfruit, jak, jack |
| 963 | pizza, pizza pie |

## F   DETAILS OF THE USER STUDY

We employed a crowdsourcing approach by hiring five annotators to determine whether each adversarial example preserves the original concept. Since our concepts are all concrete objects, we used the term "same item" to convey the notion of "same concept" in a straightforward manner. Following the user study methods of Song et al. (2018) and Zhang et al. (2024b), all five annotators voted on whether they believed each adversarial example still represented the original concept.

**Annotator Instructions.**

> **In this study, you will see two images and be asked whether they show the "same item." Please follow these guidelines when making your judgment:**
>
> 1. **Shape/Form**
>    - If the overall shape (including any accessories) in both images closely matches or approximates each other, answer "*Yes*."
>    - If the object in one image appears excessively distorted or deformed compared to the other, answer "*No*."
> 2. **Accessories**
>    - Even if the second image has additional or fewer accessories, as long as it essentially represents the same item, answer "*Yes*."
> 3. **Color**
>    - If there is no significant difference in color between the two images, answer "*Yes*."
>    - If there is a clear and noticeable color difference that affects recognizing the item, answer "*No*."

Figure 4 shows the user interface for this study.

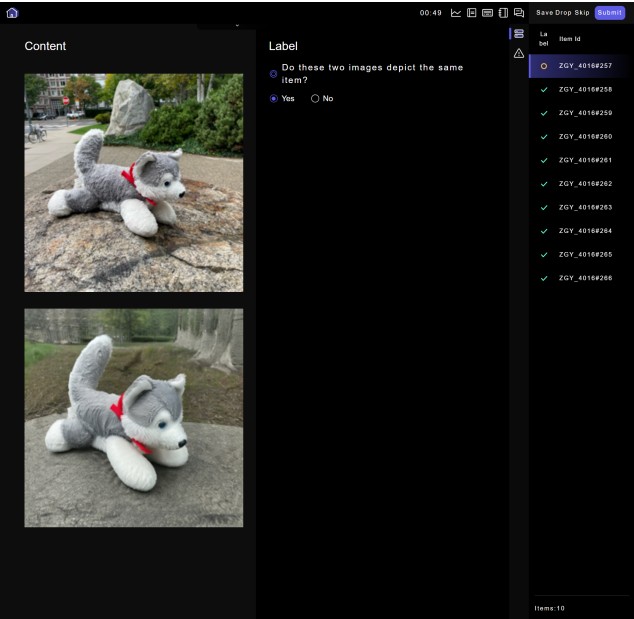

Figure 4: Screenshot of the user interface for user study.

## G  DETAILED QUALITATIVE ANALYSIS

Due to space constraints, the qualitative comparison figures in the main text are relatively small. Therefore, in this section, we present enlarged qualitative comparisons between DiffAttack and our approach. We focus on comparing DiffAttack in particular because, although it achieves a high target attack success rate, it produces lower-quality images, as shown in our user study and image quality tests. Here, we examine specific adversarial examples generated by DiffAttack to illustrate why its image quality is inferior.

- **Border Collie (first column of Figure 5).** DiffAttack removes all the dog's fur details and replaces its eyes with those of another canine, while the nose and tongue become stylized with a graffiti-like look, losing realistic details. In contrast, although our concept-based adversarial example changes the dog's pose, it preserves the animal's fur and facial details.

- **Shiny Sneakers (second column).** DiffAttack turns the sneakers into a shoe design with sharp edges, losing the smooth curves of the original model.

- **Chow Chow (third column).** DiffAttack alters all of the fur details and erases the dog's forelegs, resulting in a shape that resembles a drumstick rather than a chow chow.

- **Colorful Sneakers (fourth column).** While DiffAttack retains the purple front section and the blue rear section, it removes the yellow line in the middle and the adjacent cyan trim. Without these details, the sneaker's appearance changes to a different style altogether.

- **Cat (fifth column).** DiffAttack modifies the cat's fur patterns and gives it a distorted facial expression.

These observations further show that DiffAttack's adversarial examples degrade crucial details, resulting in a significant drop in image quality and diminished fidelity to the original concept. **To emphasize that this qualitative study is not cherry-picking, we provide the complete set of adversarial examples — both from other methods and ours — in the code repository.**

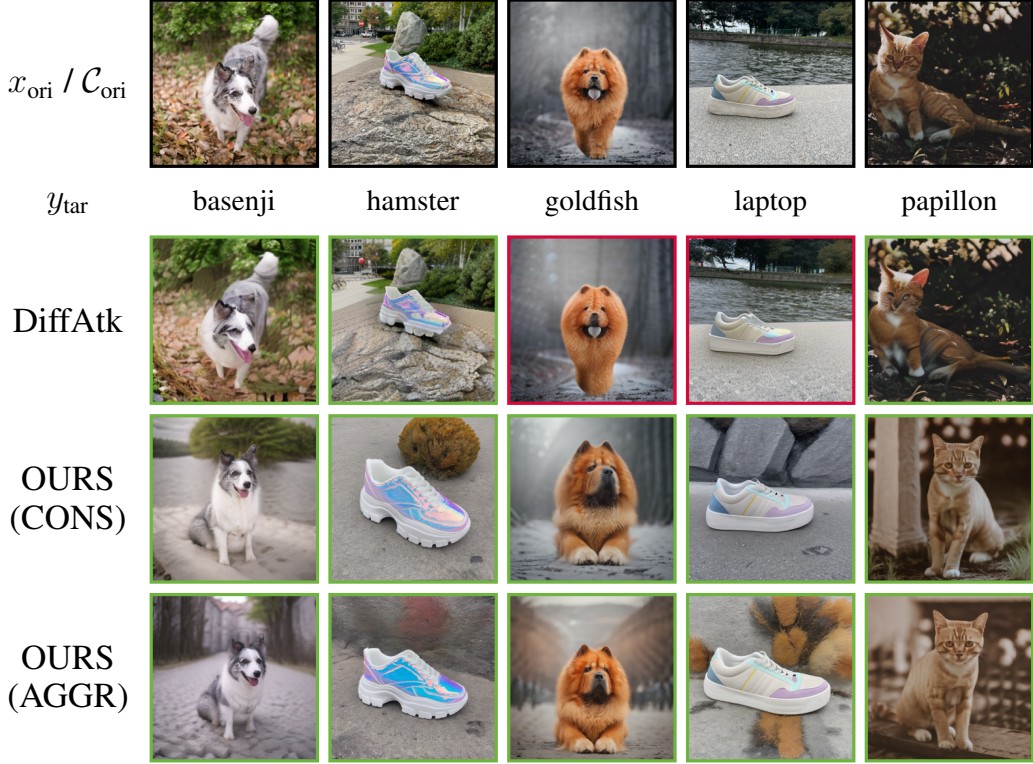

Figure 5: Qualitative comparison (zoomed in). (A **green** border indicates an example that successfully fools the classifier; **red** indicates failure.)

## H  ADDITIONAL COMMENTS ON TRANSFERABILITY

As shown in Appendix Q, our proposed method consistently achieves the best transferability among comparable approaches. However, its attack success rates are still considerably lower than those of methods explicitly optimized for transferability, such as works of Zhu et al. (2022), Wang & He (2021), Gubri et al. (2022) and Collins et al. (2025). We note that approaches targeting transferability often generate clear visual features of the target class. Especially in the setting of unrestricted adversarial attacks, directly synthesizing objects of the target class within the image is also considered valid (see Figure 5 in the appendix of Collins et al. (2025)).

From the probabilistic perspective of adversarial attack, this phenomenon is especially intuitive. As illustrated in Figure 6, $p_{\text{dis}}$ denotes the distance distribution, while $p_{\text{vic}}^{(1)}$ and $p_{\text{vic}}^{(2)}$ represent the victim distributions induced by two different classifiers. Without loss of generality, suppose adversarial examples are sampled from the product of the red distribution $p_{\text{vic}}^{(1)}$ and the blue distribution $p_{\text{dis}}$. The resulting adversarial examples concentrate in the region where these two distributions overlap. As shown in the figure, this region corresponds to low probability density under the green distribution $p_{\text{vic}}^{(2)}$, leading to poor transferability across classifiers.

Although the overall transferability of our method is relatively low, expanding $p_{\text{dis}}$ indeed brings the overlap between the blue and red distributions closer to the green distribution, making it reasonable that a larger $p_{\text{dis}}$ leads to an improvement in transferability.

However, it is important to emphasize that our goal fundamentally differs from methods explicitly designed to maximize transferability. Approaches that achieve very high transferability, such as Zhu et al. (2022) and Collins et al. (2025), typically introduce strong visual features of the target class. Doing so moves samples toward the intersection of the red and green victim distributions, thereby dramatically improving transferability. Yet this strategy comes at the cost of injecting target-

class semantics into the generated images, which directly violates our requirement of preserving the original identity-level concept.

In contrast, our method aims to investigate how enlarging the concept-based distance distribution $p_{\text{dis}}$ affects both image quality and transferability, while strictly maintaining the underlying identity. Under this constraint, extremely high transferability is not expected - and, in fact, cannot be achieved without compromising identity preservation. Although our framework could be extended to incorporate target-class features to boost transferability, we view this as beyond the scope of the current work and a promising direction for future research.

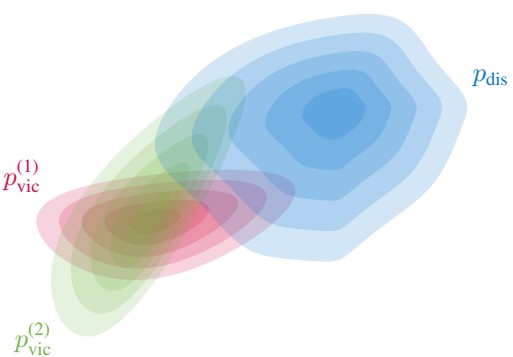

Figure 6: Transferability of adversarial attacks from a probabilistic perspective. $p_{\text{dis}}$ denotes the distance distribution, while $p_{\text{vic}}^{(1)}$ and $p_{\text{vic}}^{(2)}$ represent the victim distributions induced by two different classifiers. Without loss of generality, assume that adversarial examples are sampled from the product of the red distribution $p_{\text{vic}}^{(1)}$ and the blue distribution $p_{\text{dis}}$. In this case, the generated adversarial examples concentrate in the overlap between these two distributions. As illustrated in the figure, this overlapping region has low probability density under the green distribution $p_{\text{vic}}^{(2)}$, resulting in poor transferability. Moreover, if a sample happens to contain strong visual evidence of the target class, then both classifiers would classify it as the target with high confidence, hence the high-density regions of $p_{\text{vic}}^{(1)}$ and $p_{\text{vic}}^{(2)}$ would necessarily overlap.

## I    COMPUTE RESOURCES

All experiments are conducted on a single NVIDIA H100 Tensor Core GPU. Our method requires approximately 20 minutes per concept for concept augmentation and around 8 hours per concept for diffusion model fine-tuning. For adversarial example generation, although our method is slightly slower, it is still within the same order of magnitude.

To ensure that the diffusion model learns a high-quality identity-level concept, we train it for a long duration during the concept finetuning stage. When the concept dataset is sufficiently large (e.g., the 30 images described in the main text), the diffusion model does not collapse even after many epochs of finetuning. However, in settings like ProbAttack, where finetuning is performed on a single image, the number of epochs must be carefully controlled to avoid model collapse. For this reason, the concept finetuning time for ProbAttack is approximately 20 minutes, whereas our concept-based method requires about 8 hours.

In our experiments, we ensured that this 8-hour finetuning worked reliably across all concepts studied in the paper. In practice, the finetuning time can be reduced to some extent; however, maintaining an identity-level concept is inherently subjective and difficult to quantify, making it challenging to specify a universally optimal finetuning duration.

Table 8 summarizes the time consumption of different methods across concept augmentation, concept fine-tuning, and adversarial example generation stages.

Table 8: Time consumption of each method across different stages.

|  | NCF | ACA | DiffAttack | ProbAttack | OURS |
|---|---|---|---|---|---|
| Concept augmentation (per concept) | - | - | - | - | 20m |
| Concept finetune (per concept) | - | - | - | 20m | 8h |
| Adversarial example generation (per image) | 30s | 20s | 12s | 75s | 75s |

## J  DATASETS AND LICENSES

In this work, we use the DreamBooth dataset, which is licensed under the Creative Commons Attribution 4.0 International license.

## K  PRACTICAL USAGE GUIDELINES AND EXAMPLE SCENARIOS

### K.1  GUIDELINES

Although the term "concept" is difficult to define precisely, our method provides a clear definition: a concept can be specified by either a set $\mathcal{C}_{\text{ori}}$ or a probabilistic model. Users can therefore construct $\mathcal{C}_{\text{ori}}$ to include whatever concept variations they desire. In our experiments, we demonstrate a broad range of concept variations — background, pose, and viewpoint — leading to a highly diverse $\mathcal{C}_{\text{ori}}$ and scenarios like the one shown in Figure 1 (right).

In practice, constraints may prevent such extensive concept variations. For example, one might need to fix the background and viewpoint, leaving only the object's pose or special variations (e.g., dressing a dog in different outfits). However, if the concept variations are too limited, the intersection between $p_{\text{dis}}$ and $p_{\text{vic}}$ may be insufficient, making it harder to generate adversarial samples. Practitioners should be mindful of this trade-off.

### K.2  EXAMPLE SCENARIOS

#### K.2.1  SCENARIO 1: PROHIBITED ITEM ADVERTISEMENTS ON SOCIAL PLATFORMS

Social or second-hand platforms often employ basic classifiers to preliminarily filter user-uploaded content for prohibited items (e.g., firearms, knives, protected animals). Malicious actors aim to sell prohibited items such as a specific brand and model of firearm, a particular knife, or a specific protected animal cub. They prefer to upload images capturing all detailed features of the prohibited items (i.e., preserving concept/identity, such as the specific firearm model or exact animal cub) to attract precise target buyers while bypassing platform moderation (untargeted attack).

In such cases, preserving the concept and identity is crucial to attracting potential customers, while background and perspective variations in images are insignificant. This perfectly aligns with the applicability of concept-based adversarial attacks.

Considering the potential societal harm of this scenario, as previously discussed, social platforms should implement multiple detection systems, including AI-based detection, to prevent such attacks.

#### K.2.2  SCENARIO 2: IMPERCEPTIBLE ADVERSARIAL PATCHES IN REAL-WORLD SCENARIOS (ADVERSARIAL PATCH T-SHIRTS)

In practice, adversarial examples may be printed as patches. Early adversarial patches, though able to evade classifiers or detectors, often appeared unnatural or suspicious, making them easily noticeable by humans and limiting their effectiveness. Creating imperceptible adversarial patches remains a significant challenge.

Our method addresses this challenge by adopting brand images, logos, or cartoon characters as the concept. By altering the background or viewpoint of these concepts, we create realistic adversarial patches suitable for real-world applications.

More concretely, as demonstrated by Wang et al. (2024), adversarial examples can be created as printed patches on T-shirts to deceive detection systems. However, unnatural or suspicious patches would prompt humans to comment, "You're wearing a strange T-shirt," or, "The logo on your T-shirt looks odd." In such scenarios, our concept-based adversarial attack excels by preserving logos, branding, or cartoon imagery while subtly changing the background or making the characters perform specific actions, resulting in adversarial patches that are difficult for humans to detect.

Specifically referencing Wang et al. (2024)'s work, their algorithm primarily focuses on single-image adversarial patch creation, leading to unnatural-looking printed watermarks on T-shirts. In contrast, our concept-based adversarial attack provides a better solution.

Given the social harm posed by these attacks, real-world detection systems should employ multi-layered detection with varying thresholds and multiple scales of analysis to prevent such vulnerabilities.

## L  NEGATIVE SOCIAL EFFECT AND MITIGATION STRATEGIES

Mitigating the risks of our proposed attack is a crucial responsibility for both the machine learning community and society. Potential strategies include:

- Adversarial training using concept-based adversarial examples is a direct and general mitigation strategy. However, this may come at the cost of reduced baseline model accuracy.
- Given that our adversarial examples are directly generated from probabilistic generative models, contemporary AI-generated content detection techniques (open-source/commercial), such as frequency-domain analysis, heatmap analysis, anomaly detection, and counterfactual detection, can serve as effective countermeasures.
- In practical engineering scenarios, combining various methods according to specific application needs can significantly mitigate the threat posed by this attack.

The field of adversarial attacks continually evolves through the ongoing advancement of both attack and defense strategies. We hope our novel attack method draws sufficient attention from the community to further enhance AI Safety research.

## M  BROADER IMPACTS OF THIS WORK

This study introduces concept-based adversarial attacks, providing valuable insights into the vulnerabilities of sophisticated classifiers.

On the positive side, our work exposes critical weaknesses in systems previously considered robust, highlighting the need for enhanced security measures in classifier design. By identifying these vulnerabilities, we contribute to the development of more resilient artificial intelligence systems.

However, we acknowledge potential negative implications. The concept-based attack methods described could be misappropriated by malicious actors, for example, for identification purposes. We emphasize the importance of developing countermeasures against such exploitation and encourage the research community to consider ethical implications when building upon this work.

## N  LLM DISCLAIMER

In this work, we use large language models solely for text polishing.

## O  LIMITATIONS

Our approach is slower than similar methods because it requires time-consuming fine-tuning on a concept dataset $\mathcal{C}_{ori}$, which may also need to be built or expanded if unavailable. However, in adversarial attack scenarios, even one successful example can cause severe damage, highlighting the practical importance of our method.

| Method | Sampler / Solver | # Sampling Steps | # Attack Steps | Step Size | Notes |
|---|---|---|---|---|---|
| DiffAttack | DDIM | 50 | 30 | 0.01 | – |
| ACA | DDIM | 50 | 10 | 0.04 | – |
| NCF | – | – | 15 | 0.013 | # Color Sampling = 10 |
| ProbAttack | DDPM | 250 | – | – | $|C_{\text{ori}}| = 1$, $M = 10$, $c = 30$. |
| Concept-based | DDPM | 250 | – | – | $|C_{\text{ori}}| = 30$, $M = 10$, $c = 30$. |

Table 9: Hyperparameter settings for all compared methods. ProbAttack and our concept-based adversarial attack both use the standard stochastic diffusion (DDPM) sampler with classifier-guided Langevin dynamics, not DDIM. All other parameters are at default.

## P  BASELINE SETTINGS

We list the hyperparameter settings for all compared methods in Table 9. Note that DiffAttack/ACA and ProbAttack/Concept-based Attack rely on fundamentally different attack mechanisms: the former are based on DDIM and operate in the latent space, while the latter directly sample from $p_{\text{adv}}$. As a result, the number of sampling steps is not directly comparable across these methods, and the latter do not require any additional attack steps.

## Q  FULL MAIN EXPERIMENTAL RESULTS

Due to space constraints, the main text reports only the white-box Top-1 and black-box Top-5 results using ResNet-50 as the surrogate classifier. In this section, we provide the full set of experimental results, including targeted attack success rates (white-box Top-1, black-box Top-1, Top-5, Top-10, and Top-100) in Section Q.1, and the similarity and image quality evaluations in Section Q.2.

### Q.1  TARGETED ATTACK SUCCESS RATES

Table 10: Attack success rates (%) on ImageNet classifiers, with ResNet-50 serving as the white-box victim (surrogate) classifier.

| | NCF | ACA | DiffAttack | ProbAttack | OURS (CONS) | OURS (AGGR) |
|---|---|---|---|---|---|---|
| **White-box** | | | **Targeted-Top1** | | | |
| ResNet 50 | 1.15 | 6.03 | 84.23 | 59.23 | **97.82** | **97.82** |
| **Transferability** | | | | | | |
| VGG19 | 0.26 | 0.26 | **1.67** | 0.38 | 0.00 | 1.15 |
| ResNet 152 | 0.13 | 0.38 | **1.79** | 0.77 | 0.26 | **1.79** |
| DenseNet 161 | 0.00 | 0.26 | 1.67 | 0.51 | 0.26 | **2.82** |
| Inception V3 | 0.13 | 0.13 | 0.90 | 0.51 | 0.00 | **1.54** |
| EfficientNet B7 | 0.00 | 0.26 | 0.38 | 0.00 | 0.13 | **1.15** |
| **Adversarial Defence** | | | | | | |
| ResNet 50 Adv | 0.00 | 0.38 | 0.77 | 0.26 | 0.00 | **1.15** |
| Inception V3 Adv | 0.00 | 0.26 | **1.03** | 0.26 | 0.00 | 1.03 |
| EfficientNet B7 Adv | 0.00 | 0.51 | 0.64 | 0.38 | 0.26 | **1.41** |
| Ensemble IncRes V2 | 0.00 | 0.26 | 0.38 | 0.64 | 0.00 | **1.28** |
| **White-box** | | | **Targeted-Top5** | | | |
| ResNet 50 | 3.21 | 10.64 | 90.64 | 72.82 | **99.87** | **99.87** |
| **Transferability** | | | | | | |
| VGG19 | 1.28 | 1.67 | **4.36** | 2.44 | 2.05 | **4.36** |
| ResNet 152 | 1.41 | 1.92 | 8.33 | 3.33 | 2.82 | **8.72** |
| DenseNet 161 | 1.41 | 2.05 | 7.44 | 3.97 | 3.85 | **11.54** |
| Inception V3 | 0.90 | 1.41 | 3.08 | 2.56 | 1.28 | **4.74** |
| EfficientNet B7 | 1.41 | 1.67 | 1.79 | 1.41 | 1.28 | **3.97** |
| **Adversarial Defence** | | | | | | |
| ResNet 50 Adv | 0.90 | 1.15 | 3.46 | 2.56 | 1.79 | **5.64** |
| Inception V3 Adv | 1.15 | 1.28 | 3.21 | 2.18 | 0.90 | **3.72** |
| EfficientNet B7 Adv | 0.26 | 1.15 | 2.05 | 2.31 | 1.67 | **6.41** |
| Ensemble IncRes V2 | 0.77 | 1.28 | 2.69 | 1.92 | 0.77 | **5.00** |
| **White-box** | | | **Targeted-Top10** | | | |
| ResNet 50 | 4.23 | 12.69 | 93.46 | 75.64 | **99.87** | **99.87** |
| **Transferability** | | | | | | |
| VGG19 | 2.69 | 2.69 | **8.21** | 3.46 | 3.21 | 6.79 |
| ResNet 152 | 2.44 | 3.59 | 14.74 | 5.77 | 6.28 | **15.13** |
| DenseNet 161 | 2.31 | 3.72 | 12.69 | 6.15 | 8.72 | **19.62** |
| Inception V3 | 1.28 | 2.31 | 5.13 | 3.46 | 2.31 | **6.54** |
| EfficientNet B7 | 2.31 | 2.82 | 4.23 | 3.21 | 3.59 | **6.54** |
| **Adversarial Defence** | | | | | | |
| ResNet 50 Adv | 1.41 | 1.92 | 5.64 | 3.59 | 2.69 | **8.46** |
| Inception V3 Adv | 1.79 | 1.79 | 5.13 | 3.46 | 2.31 | **6.54** |
| EfficientNet B7 Adv | 0.38 | 2.05 | 3.08 | 3.46 | 2.44 | **9.36** |
| Ensemble IncRes V2 | 1.28 | 2.05 | 3.85 | 3.08 | 1.79 | **7.18** |

Table 11: Attack success rates (%) on ImageNet classifiers, with MobileNet v2 serving as the white-box victim (surrogate) classifier.

| | NCF | ACA | DiffAttack | ProbAttack | OURS (CONS) | OURS (AGGR) |
|---|---|---|---|---|---|---|
| **White-box** | | | | **Targeted-Top1** | | |
| MN-V2 | 3.85 | 12.95 | 91.92 | 51.15 | **97.31** | **97.31** |
| **Transferability** | | | | | | |
| VGG19 | 0.00 | 0.13 | **1.79** | 0.00 | 0.00 | 1.54 |
| ResNet 152 | 0.13 | 0.26 | 0.90 | 0.13 | 0.00 | **1.03** |
| DenseNet 161 | 0.00 | 0.26 | 0.90 | 1.03 | 0.00 | **1.28** |
| Inception V3 | 0.00 | 0.13 | 0.51 | 0.38 | 0.00 | **0.90** |
| EfficientNet B7 | 0.00 | 0.00 | 0.64 | 0.13 | 0.13 | **1.28** |
| **Adversarial Defence** | | | | | | |
| Inception V3 Adv | 0.00 | 0.26 | 0.77 | 0.13 | 0.13 | **1.15** |
| EfficientNet B7 Adv | 0.00 | 0.26 | 0.26 | 0.26 | 0.38 | **1.41** |
| Ensemble IncRes V2 | 0.00 | 0.38 | 0.77 | 0.26 | 0.13 | **1.28** |
| **White-box** | | | | **Targeted-Top5** | | |
| MN-V2 | 6.41 | 17.18 | 95.64 | 65.64 | **99.74** | **99.74** |
| **Transferability** | | | | | | |
| VGG19 | 1.03 | 2.05 | 4.36 | 1.54 | 1.92 | **4.74** |
| ResNet 152 | 1.15 | 1.54 | 3.72 | 1.79 | 1.92 | **4.87** |
| DenseNet 161 | 1.15 | 1.54 | 5.51 | 3.46 | 2.56 | **7.18** |
| Inception V3 | 0.90 | 1.15 | 2.44 | 1.92 | 1.79 | **3.97** |
| EfficientNet B7 | 1.03 | 1.41 | 2.05 | 0.51 | 0.90 | **4.36** |
| **Adversarial Defence** | | | | | | |
| Inception V3 Adv | 0.90 | 1.15 | 2.56 | 1.67 | 0.90 | **3.08** |
| EfficientNet B7 Adv | 0.38 | 1.15 | 1.92 | 1.54 | 1.41 | **5.77** |
| Ensemble IncRes V2 | 0.38 | 1.67 | 2.56 | 1.15 | 0.90 | **4.36** |
| **White-box** | | | | **Targeted-Top10** | | |
| MN-V2 | 8.33 | 20.00 | 97.05 | 72.31 | **99.74** | **99.74** |
| **Transferability** | | | | | | |
| VGG19 | 1.28 | 2.95 | **7.31** | 2.69 | 3.33 | 6.92 |
| ResNet 152 | 1.54 | 2.56 | 7.05 | 3.59 | 2.82 | **8.08** |
| DenseNet 161 | 1.67 | 3.08 | 9.10 | 5.00 | 4.36 | **12.44** |
| Inception V3 | 1.41 | 2.82 | 4.23 | 2.82 | 2.82 | **5.90** |
| EfficientNet B7 | 1.79 | 3.33 | 3.72 | 2.18 | 3.08 | **7.31** |
| **Adversarial Defence** | | | | | | |
| Inception V3 Adv | 1.28 | 1.41 | 4.10 | 2.56 | 2.05 | **5.90** |
| EfficientNet B7 Adv | 0.64 | 2.44 | 3.59 | 2.82 | 3.97 | **8.72** |
| Ensemble IncRes V2 | 0.77 | 2.31 | 3.72 | 2.95 | 2.31 | **7.18** |

Table 12: Attack success rates (%) on ImageNet classifiers, with ViT-Base serving as the white-box victim (surrogate) classifier.

| | NCF | ACA | DiffAttack | ProbAttack | OURS (CONS) | OURS (AGGR) |
|---|---|---|---|---|---|---|
| **White-box** | | | | Targeted-Top1 | | |
| ViT-B | 0.90 | 3.46 | 81.41 | 63.85 | **85.51** | **85.51** |
| **Transferability** | | | | | | |
| VGG19 | 0.13 | 0.26 | 0.51 | 0.13 | 0.13 | **1.28** |
| ResNet 152 | 0.00 | 0.38 | 0.77 | 0.00 | 0.00 | **1.41** |
| DenseNet 161 | 0.00 | 0.26 | **1.15** | 0.51 | 0.00 | 1.15 |
| Inception V3 | 0.00 | 0.38 | 0.38 | 0.13 | 0.13 | **0.64** |
| EfficientNet B7 | 0.00 | 0.26 | 0.64 | 0.00 | 0.00 | **0.77** |
| **Adversarial Defence** | | | | | | |
| Inception V3 Adv | 0.00 | 0.26 | 0.38 | 0.13 | 0.13 | **0.90** |
| EfficientNet B7 Adv | 0.00 | 0.38 | 0.90 | 0.51 | 0.51 | **1.15** |
| Ensemble IncRes V2 | 0.00 | 0.13 | 1.03 | 0.13 | 0.00 | **1.15** |
| **White-box** | | | | Targeted-Top5 | | |
| ViT-B | 2.44 | 6.67 | 92.44 | 89.74 | **94.87** | **94.87** |
| **Transferability** | | | | | | |
| VGG19 | 1.03 | 1.41 | 1.79 | 1.28 | 1.41 | **3.33** |
| ResNet 152 | 0.64 | 1.67 | 3.33 | 1.03 | 1.67 | **4.74** |
| DenseNet 161 | 0.64 | 1.67 | 4.87 | 2.18 | 1.67 | **5.51** |
| Inception V3 | 0.77 | 1.41 | 2.95 | 1.54 | 0.90 | **3.08** |
| EfficientNet B7 | 0.90 | 1.92 | 2.69 | 1.28 | 1.15 | **3.08** |
| **Adversarial Defence** | | | | | | |
| Inception V3 Adv | 0.51 | 1.92 | 3.21 | 0.64 | 0.90 | **3.46** |
| EfficientNet B7 Adv | 0.26 | 1.79 | 3.08 | 1.67 | 1.41 | **3.85** |
| Ensemble IncRes V2 | 0.51 | 1.41 | 3.08 | 1.28 | 0.90 | **3.21** |
| **White-box** | | | | Targeted-Top10 | | |
| ViT-B | 3.46 | 8.97 | 95.13 | 92.05 | **95.51** | **95.51** |
| **Transferability** | | | | | | |
| VGG19 | 1.79 | 2.69 | 3.46 | 2.05 | 2.82 | **4.49** |
| ResNet 152 | 1.28 | 1.92 | 5.77 | 2.05 | 2.82 | **7.69** |
| DenseNet 161 | 1.15 | 2.82 | 6.92 | 3.08 | 4.23 | **8.59** |
| Inception V3 | 1.28 | 2.05 | 4.87 | 1.79 | 1.92 | **5.00** |
| EfficientNet B7 | 1.54 | 2.95 | 4.49 | 2.44 | 3.21 | **5.64** |
| **Adversarial Defence** | | | | | | |
| Inception V3 Adv | 0.77 | 2.18 | 5.13 | 2.44 | 2.05 | **5.26** |
| EfficientNet B7 Adv | 0.38 | 2.31 | 5.51 | 2.82 | 3.33 | **6.03** |
| Ensemble IncRes V2 | 0.90 | 2.31 | 4.87 | 2.31 | 2.56 | **5.00** |

Table 13: Attack success rates (%) on ImageNet classifiers, with ConvNext serving as the white-box victim (surrogate) classifier.

| | NCF | ACA | DiffAttack | ProbAttack | OURS (CONS) | OURS (AGGR) |
|---|---|---|---|---|---|---|
| **White-box** | | | | **Targeted-Top1** | | |
| ConvNext | 1.03 | 5.38 | 83.59 | 60.38 | **94.74** | **94.74** |
| **Transferability** | | | | | | |
| VGG19 | 0.00 | 0.26 | **1.28** | 0.26 | 0.00 | 1.15 |
| ResNet 152 | 0.00 | 0.38 | 1.54 | 0.51 | 0.13 | **1.67** |
| DenseNet 161 | 0.00 | 0.26 | 1.54 | 0.51 | 0.13 | **2.31** |
| Inception V3 | 0.13 | 0.26 | 0.77 | 0.38 | 0.00 | **1.28** |
| EfficientNet B7 | 0.00 | 0.26 | 0.51 | 0.00 | 0.13 | **1.03** |
| **Adversarial Defence** | | | | | | |
| Inception V3 Adv | 0.00 | 0.26 | 0.90 | 0.26 | 0.00 | **1.03** |
| EfficientNet B7 Adv | 0.00 | 0.51 | 0.77 | 0.38 | 0.38 | **1.28** |
| Ensemble IncRes V2 | 0.00 | 0.26 | 0.51 | 0.51 | 0.00 | **1.28** |
| **White-box** | | | | **Targeted-Top5** | | |
| ConvNext | 2.95 | 9.36 | 91.15 | 78.85 | **98.59** | **98.59** |
| **Transferability-Top5** | | | | | | |
| VGG19 | 1.15 | 1.54 | 3.46 | 2.05 | 1.79 | **4.10** |
| ResNet 152 | 1.15 | 1.79 | 6.92 | 2.69 | 2.44 | **7.56** |
| DenseNet 161 | 1.15 | 1.92 | 6.67 | 3.33 | 3.21 | **9.62** |
| Inception V3 | 0.90 | 1.41 | 3.08 | 2.18 | 1.15 | **4.23** |
| EfficientNet B7 | 1.28 | 1.79 | 2.05 | 1.41 | 1.28 | **3.72** |
| **Adversarial Defence** | | | | | | |
| Inception V3 Adv | 0.90 | 1.41 | 3.21 | 1.67 | 0.90 | **3.59** |
| EfficientNet B7 Adv | 0.26 | 1.28 | 2.44 | 2.05 | 1.54 | **5.51** |
| Ensemble IncRes V2 | 0.64 | 1.28 | 2.82 | 1.67 | 0.77 | **4.49** |
| **White-box** | | | | **Targeted-Top10** | | |
| ConvNext | 3.97 | 11.67 | 93.97 | 81.54 | **98.72** | **98.72** |
| **Transferability** | | | | | | |
| VGG19 | 2.44 | 2.69 | **6.67** | 2.95 | 3.08 | 6.15 |
| ResNet 152 | 2.05 | 3.08 | 11.92 | 4.49 | 5.13 | **12.82** |
| DenseNet 161 | 1.92 | 3.46 | 10.90 | 5.13 | 7.18 | **16.41** |
| Inception V3 | 1.28 | 2.18 | 5.00 | 2.82 | 2.18 | **6.15** |
| EfficientNet B7 | 2.05 | 2.82 | 4.36 | 2.95 | 3.46 | **6.28** |
| **Adversarial Defence** | | | | | | |
| Inception V3 Adv | 1.41 | 1.92 | 5.13 | 3.08 | 2.18 | **6.15** |
| EfficientNet B7 Adv | 0.38 | 2.18 | 3.72 | 3.21 | 2.69 | **8.21** |
| Ensemble IncRes V2 | 1.15 | 2.18 | 4.23 | 2.82 | 2.05 | **6.54** |

Table 14: Attack success rates (%) on ImageNet classifiers, with ResNet-50 Adv serving as the white-box victim (surrogate) classifier.

| | NCF | ACA | DiffAttack | ProbAttack | OURS (CONS) | OURS (AGGR) |
|---|---|---|---|---|---|---|
| **White-box** | | | | **Targeted-Top1** | | |
| ResNet-50 Adv | 0.90 | 5.13 | 81.79 | 61.28 | **94.23** | **94.23** |
| **Transferability** | | | | | | |
| VGG19 | 0.13 | 0.38 | 1.67 | 0.51 | 0.00 | **2.31** |
| ResNet 152 | 0.00 | 0.51 | **2.56** | 1.03 | 0.38 | **2.56** |
| DenseNet 161 | 0.00 | 0.38 | 2.31 | 0.77 | 0.38 | **3.97** |
| Inception V3 | 0.00 | 0.26 | 1.28 | 0.77 | 0.00 | **2.18** |
| EfficientNet B7 | 0.00 | 0.38 | 0.51 | 0.00 | 0.26 | **1.67** |
| **Adversarial Defence** | | | | | | |
| Inception V3 Adv | 0.00 | 0.51 | 1.67 | 0.51 | 0.00 | **1.79** |
| EfficientNet B7 Adv | 0.00 | 0.90 | 1.15 | 0.64 | 0.51 | **2.56** |
| Ensemble IncRes V2 | 0.00 | 0.51 | 0.64 | 1.15 | 0.00 | **2.31** |
| **White-box** | | | | **Targeted-Top5** | | |
| ResNet-50 Adv | 2.82 | 9.62 | 89.74 | 75.00 | **98.08** | **98.08** |
| **Transferability** | | | | | | |
| VGG19 | 1.03 | 2.31 | 5.77 | 3.33 | 2.82 | **6.15** |
| ResNet 152 | 1.15 | 2.69 | 11.28 | 4.49 | 3.85 | **11.79** |
| DenseNet 161 | 1.15 | 2.82 | 10.00 | 5.38 | 5.26 | **15.38** |
| Inception V3 | 0.77 | 1.92 | 4.36 | 3.46 | 1.79 | **6.67** |
| EfficientNet B7 | 1.15 | 2.31 | 2.56 | 1.92 | 1.79 | **5.51** |
| **Adversarial Defence** | | | | | | |
| Inception V3 Adv | 0.90 | 1.92 | 5.13 | 3.46 | 1.41 | **6.15** |
| EfficientNet B7 Adv | 0.26 | 1.79 | 3.33 | 3.59 | 2.56 | **10.00** |
| Ensemble IncRes V2 | 0.64 | 1.92 | 4.36 | 3.08 | 1.15 | **7.95** |
| **White-box** | | | | **Targeted-Top10** | | |
| ResNet-50 Adv | 3.85 | 11.67 | 92.95 | 77.56 | **98.97** | **98.97** |
| **Transferability** | | | | | | |
| VGG19 | 2.18 | 3.72 | 8.97 | 4.62 | 4.36 | **10.77** |
| ResNet 152 | 2.05 | 4.87 | 18.97 | 7.44 | 8.08 | **19.49** |
| DenseNet 161 | 1.92 | 5.00 | 16.28 | 7.95 | 11.15 | **25.13** |
| Inception V3 | 1.03 | 3.21 | 6.92 | 4.62 | 3.08 | **8.72** |
| EfficientNet B7 | 1.92 | 3.85 | 5.64 | 4.36 | 4.87 | **8.72** |
| **Adversarial Defence** | | | | | | |
| Inception V3 Adv | 1.41 | 2.69 | 8.21 | 5.38 | 3.59 | **10.51** |
| EfficientNet B7 Adv | 0.38 | 3.08 | 5.00 | 5.38 | 3.85 | **14.74** |
| Ensemble IncRes V2 | 1.03 | 3.08 | 6.15 | 4.74 | 2.82 | **11.41** |

## Q.2 SIMILARITY AND IMAGE QUALITY

Due to cost constraints, only the ResNet-50 results reported in the main text include the user study.

Table 15: Quantitative comparison of similarity to the original images and no reference image quality metrics for unrestricted adversarial examples with ResNet-50 serving as the victim (surrogate) classifier.

|  | Clean | NCF | ACA | DiffAttack | ProbAttack | OURS (CONS) | OURS (AGGR) |
|---|---|---|---|---|---|---|---|
| **Similarity** | | | | | | | |
| ↑ User Study | N/A | 0.1859 | 0.2808 | 0.7577 | 0.8041 | **0.9654** | 0.8808 |
| ↑ Avg. Clip Score | 1.0 | **0.8728** | 0.7861 | 0.8093 | 0.8581 | 0.8283 | 0.8043 |
| **Image Quality** | | | | | | | |
| ↑ HyperIQA | 0.7255 | 0.5075 | 0.6462 | 0.5551 | 0.6675 | **0.6947** | 0.6809 |
| ↑ DBCNN | 0.6956 | 0.5096 | 0.6103 | 0.5294 | 0.6161 | **0.6572** | 0.6399 |
| ↑ ARNIQA | 0.7667 | 0.5978 | 0.6879 | 0.6909 | 0.7009 | **0.7335** | 0.7154 |
| ↑ MUSIQ-AVA | 4.3760 | 3.8135 | 4.2687 | 4.3130 | 4.3130 | **4.5305** | 4.5250 |
| ↑ NIMA-AVA | 4.5595 | 3.7916 | 4.4511 | 4.0589 | 4.5168 | **4.7575** | 4.7401 |
| ↑ MUSIQ-KonIQ | 65.0549 | 50.5022 | 59.0840 | 52.5399 | 58.1563 | **63.7486** | 62.2217 |
| ↑ TReS | 93.2127 | 64.7050 | 85.8435 | 74.1167 | 84.3131 | **90.4488** | 88.0836 |

Table 16: Quantitative comparison of similarity to the original images and no reference image quality metrics for unrestricted adversarial examples with MN-V2 serving as the victim (surrogate) classifier.

|  | Clean | NCF | ACA | DiffAttack | ProbAttack | OURS (CONS) | OURS (AGGR) |
|---|---|---|---|---|---|---|---|
| **Similarity** | | | | | | | |
| ↑ Avg. Clip Score | 1.0 | **0.8783** | 0.7756 | 0.8197 | 0.8693 | 0.8229 | 0.7988 |
| **Image Quality** | | | | | | | |
| ↑ HyperIQA | 0.7255 | 0.5002 | 0.6486 | 0.5471 | 0.6808 | **0.6998** | 0.6865 |
| ↑ DBCNN | 0.6956 | 0.5040 | 0.6175 | 0.5236 | 0.6310 | **0.6577** | 0.6402 |
| ↑ ARNIQA | 0.7667 | 0.5972 | 0.6909 | 0.6872 | 0.7112 | **0.7328** | 0.7142 |
| ↑ MUSIQ-AVA | 4.3760 | 3.7912 | 4.3042 | 4.0496 | 4.3236 | **4.6125** | 4.5700 |
| ↑ NIMA-AVA | 4.5595 | 3.7879 | 4.4532 | 4.0549 | 4.5317 | **4.8172** | 4.7778 |
| ↑ MUSIQ-KonIQ | 65.0549 | 49.9072 | 59.5258 | 51.9209 | 59.7813 | **63.5363** | 61.6641 |
| ↑ TReS | 93.2127 | 63.7643 | 86.2644 | 73.0392 | 86.4376 | **90.7757** | 88.9947 |

Table 17: Quantitative comparison of similarity to the original images and no reference image quality metrics for unrestricted adversarial examples with ViT-Base serving as the victim (surrogate) classifier.

|  | Clean | NCF | ACA | DiffAttack | ProbAttack | OURS (CONS) | OURS (AGGR) |
|---|---|---|---|---|---|---|---|
| **Similarity** | | | | | | | |
| ↑ Avg. Clip Score | 1.0 | **0.8733** | 0.7681 | 0.8040 | 0.8586 | 0.8222 | 0.8104 |
| **Image Quality** | | | | | | | |
| ↑ HyperIQA | 0.7255 | 0.5006 | 0.6324 | 0.5475 | 0.6796 | **0.7077** | 0.6961 |
| ↑ DBCNN | 0.6956 | 0.5027 | 0.5842 | 0.5222 | 0.6254 | **0.6652** | 0.6526 |
| ↑ ARNIQA | 0.7667 | 0.5879 | 0.6765 | 0.7120 | 0.7120 | **0.7351** | 0.7195 |
| ↑ MUSIQ-AVA | 4.3760 | 3.7822 | 4.3131 | 4.0400 | 4.2450 | **4.5401** | 4.5345 |
| ↑ NIMA-AVA | 4.5595 | 3.7666 | 4.4775 | 4.0321 | 4.4918 | 4.7706 | **4.7747** |
| ↑ MUSIQ-KonIQ | 65.0549 | 50.0413 | 57.6105 | 52.0604 | 59.3605 | **64.2536** | 62.4264 |
| ↑ TReS | 93.2127 | 64.1109 | 83.6834 | 73.4362 | 85.8749 | **91.7463** | 89.7815 |

Table 18: Quantitative comparison of similarity to the original images and no reference image quality metrics for unrestricted adversarial examples with ConvNeXT serving as the victim (surrogate) classifier.

| | Clean | NCF | ACA | DiffAttack | ProbAttack | OURS (CONS) | OURS (AGGR) |
|---|---|---|---|---|---|---|---|
| **Similarity** | | | | | | | |
| ↑ Avg. Clip Score | 1.0 | **0.8621** | 0.7542 | 0.7964 | 0.8467 | 0.8139 | 0.8011 |
| **Image Quality** | | | | | | | |
| ↑ HyperIQA | 0.7255 | 0.4928 | 0.6281 | 0.5392 | 0.6703 | **0.6933** | 0.6810 |
| ↑ DBCNN | 0.6956 | 0.4951 | 0.5784 | 0.5179 | 0.6201 | **0.6531** | 0.6405 |
| ↑ ARNIQA | 0.7667 | 0.5835 | 0.6710 | 0.6801 | 0.7066 | **0.7294** | 0.7118 |
| ↑ MUSIQ-AVA | 4.3760 | 3.7511 | 4.2814 | 4.0206 | 4.2314 | **4.5116** | 4.5030 |
| ↑ NIMA-AVA | 4.5595 | 3.7524 | 4.4433 | 4.0215 | 4.4830 | 4.7464 | **4.7492** |
| ↑ MUSIQ-KonIQ | 65.0549 | 49.3812 | 57.0114 | 51.7324 | 58.9012 | **63.1025** | 61.3571 |
| ↑ TReS | 93.2127 | 62.9011 | 82.7410 | 72.1109 | 85.2442 | **90.9211** | 88.7724 |

Table 19: Quantitative comparison of similarity to the original images and no reference image quality metrics for unrestricted adversarial examples with ResNet-50 Adv. serving as the victim (surrogate) classifier.

| | Clean | NCF | ACA | DiffAttack | ProbAttack | OURS (CONS) | OURS (AGGR) |
|---|---|---|---|---|---|---|---|
| **Similarity** | | | | | | | |
| ↑ Avg. Clip Score | 1.0 | **0.8556** | 0.7485 | 0.7877 | 0.8367 | 0.8059 | 0.7945 |
| **Image Quality** | | | | | | | |
| ↑ HyperIQA | 0.7255 | 0.4854 | 0.6221 | 0.5297 | 0.6617 | **0.6875** | 0.6750 |
| ↑ DBCNN | 0.6956 | 0.4908 | 0.5718 | 0.5070 | 0.6078 | **0.6473** | 0.6342 |
| ↑ ARNIQA | 0.7667 | 0.5764 | 0.6654 | 0.6683 | 0.6932 | **0.7231** | 0.7070 |
| ↑ MUSIQ-AVA | 4.3760 | 3.7435 | 4.2637 | 4.0130 | 4.2223 | **4.5031** | 4.4966 |
| ↑ NIMA-AVA | 4.5595 | 3.7453 | 4.4342 | 4.0113 | 4.4676 | **4.7399** | 4.7351 |
| ↑ MUSIQ-KonIQ | 65.0549 | 49.3740 | 57.0028 | 51.7202 | 58.1513 | **63.0946** | 61.3527 |
| ↑ TReS | 93.2127 | 62.8961 | 82.7340 | 72.1034 | 84.3081 | **90.4438** | 88.0786 |

