# OpenReview forum: "Concept-based Adversarial Attack: a Probabilistic Perspective"
_ICLR.cc/2026/Conference — ICLR 2026 Poster_

### Official Review · Reviewer_2if2 · 2025-10-27

**Soundness:** 3
**Presentation:** 2
**Contribution:** 2
**Rating:** 4
**Confidence:** 3

**Summary:**

This paper presents a novel concept-based adversarial attack framework that extends probabilistic attacks from single-image perturbations to concept-level perturbations. By generalizing the distance distribution $p_{dis}$ from an individual image to an entire concept, the method effectively reduces the divergence from the victim distribution $p_{vic}$, leading to improved attack success rates and higher-quality adversarial examples. The paper also introduces a concept augmentation strategy using modern generative models to enhance distributional diversity, and demonstrates clear performance gains over existing probabilistic attack methods.

**Strengths:**

1. The paper presents a unified and elegant probabilistic framework that generalizes single-image attacks to concept-level attacks, treating the former as a special case of the latter and achieving a smooth transition from $x_{ori}$ to $C_{ori}$.
2. Experimental results strongly demonstrate that the proposed method outperforms existing baselines in terms of targeted attack success rate, while generating higher-quality adversarial examples that effectively preserve the original concept semantics.

**Weaknesses:**

1. The proposed work builds upon the probabilistic perspective introduced in [1] and extends it to concept-level attacks. However, concept-based or unrestricted attacks using generative models have already been explored in prior works, including GAN-based [2] and diffusion-based approaches [3,4]. In addition, several studies have examined distance metrics in unrestricted attacks [5,6]. These overlaps weaken the originality of the contribution.
2. The derivation of Theorem 1 relies on the assumption that $p_{dis}$ follows a Gaussian distribution. While this is a common simplification in theoretical analyses, the authors do not sufficiently discuss whether the theoretical guarantees still hold when using non-Gaussian PGMs (e.g., VAEs or EBMs). This raises concerns about the generality of the proposed framework.
3. The concept augmentation process depends heavily on powerful pretrained models (e.g., SDXL) and considerable computational resources (e.g., LoRA fine-tuning), which may limit the reproducibility and applicability of the method in resource-constrained settings.

**References:**

[1] Zhang, Andi, Mingtian Zhang, and Damon Wischik. "Constructing semantics-aware adversarial examples with a probabilistic perspective." NeurIPS, 2024.

[2] Xiao, Chaowei, et al. "Generating adversarial examples with adversarial networks." IJCAI, 2018.

[3] Dai, Xuelong, Kaisheng Liang, and Bin Xiao. "Advdiff: Generating unrestricted adversarial examples using diffusion models." ECCV, 2024.

[4] Chen, Xinquan, et al. "Advdiffuser: Natural adversarial example synthesis with diffusion models." ICCV, 2023.

[5] Laidlaw, Cassidy, Sahil Singla, and Soheil Feizi. "Perceptual Adversarial Robustness: Defense Against Unseen Threat Models." ICLR, 2021.

[6] Zhao, Zhengyu, Zhuoran Liu, and Martha Larson. "Towards large yet imperceptible adversarial image perturbations with perceptual color distance." CVPR, 2020.

**Questions:**

1. Please discuss the potential limitations of assuming a Gaussian distribution for $p_{dis}$ in Theorem 1. If a non-Gaussian PGM is used, can the reduction in KL divergence still be theoretically guaranteed? Is there a more general metric that could replace KL divergence to provide broader theoretical validity?
2. “Concept augmentation” plays a key role in improving diversity and performance, but it relies on large pretrained models such as SDXL with LoRA fine-tuning. Under limited computational resources, how would the quality, diversity, and effectiveness of adversarial samples degrade if simpler or smaller PGMs were used? Would the method remain effective in low-resource settings?
3. Tables 4 and 5 show that under the conservative (CONS) strategy, increasing the number of selected samples $M$ (from 1 to 10) decreases transferability while improving image quality and similarity. This suggests that the conservative strategy may prioritize perceptual quality over attack strength as $M$ increases. Please discuss how to better balance this trade-off between transferability and quality, and explain the specific reasons behind the drop in transferability for larger $M$.

---

> ### Author Response · Authors · 2025-11-21
>
> Thank you for your insightful and thought-provoking review. Our responses are provided below.
>
> ## Weaknesses 1
>
> > However, concept-based or unrestricted attacks using generative models have already been explored in prior works, including…
>
> Thank you for your suggestions. We have incorporated discussions of the referenced works into the revised version of the paper, specifically in the Related Work section.
>
> We would also like to emphasize that, to the best of our knowledge, our method is indeed the only existing approach capable of generating adversarial examples based on identity-level concepts. To make this distinction clearer, we have revised the exposition in the Related Work section and added the comparison table below to highlight this novelty.
>
> | Unrestricted Adversarial Attack Method | Single-image | Identity-level Concept | Class-level Concept |
> |----------------------------------------|--------------|------------------------|---------------------|
> | Bhattad et al. (2020) | Yes | No | No |
> | Hosseini & Poovendran (2018) | Yes | No | No |
> | Colorfool (Shamsabadi et al., 2020) | Yes | No | No |
> | Zhao et al. (2020) | Yes | No | No |
> | NCF (Yuan et al., 2022) | Yes | No | No |
> | ACA (Chen et al., 2024b) | Yes | No | No |
> | DiffAttack (Chen et al., 2024a) | Yes | No | No |
> | ProbAttack (Zhang et al., 2024b) | Yes | No | No |
> | AdvGAN (Xiao et al., 2018a) | Yes | No | No |
> | Xiao et al. (2018b) | Yes | No | No |
> | Perceptual Adv. Attack (Laidlaw et al., 2020) | Yes | No | No |
> | Song et al. (2018) | No | No | Yes |
> | NatADiff (Collins et al., 2025) | No | No | Yes |
> | AdvDiff (Dai et al., 2024) | No | No | Yes |
> | AdvDiffuser (Chen et al., 2023) | Yes | No | Yes |
> | **Ours: Concept-based Adv. Attack** | **Yes** | **Yes** | **Yes** |
>
> ## Weakness 2 & Question 1
>
> > … Gaussian distribution. While this is a common simplification in theoretical analyses, the authors do not sufficiently discuss whether the theoretical guarantees still hold when using non-Gaussian PGMs …
>
> > If a non-Gaussian PGM is used, can the reduction in KL divergence still be theoretically guaranteed?
>
> Thank you very much for pointing out that modeling the distribution as Gaussian is overly simplistic and not well aligned with modern PGMs. After reflecting on this issue, we realized that our conclusions can in fact be generalized to Gibbs distributions (of which Gaussian distributions are a special case). Accordingly, we have revised the section containing Theorem 1 as well as its proof in the appendix. Please refer to the updated version in our revision.
>
> By extending the result to Gibbs distributions, the theoretical framework becomes significantly more compatible with contemporary PGMs. For example, EBMs explicitly learn a Gibbs distribution, while modern diffusion models can be interpreted as implicitly learning the energy function of a Gibbs distribution.
>
> We sincerely appreciate your insight. Without your guidance, the theoretical component of this paper would not have reached its current level of rigor.
>
> > Is there a more general metric that could replace KL divergence to provide broader theoretical validity?
>
> Please refer to the updated revision, specifically Appendix A, for the revised proof of Theorem 1. The identity
> $$\frac{d KL(p||q)}{d \beta} =E_{X\sim p}[D(X,\mu)] -E_{X\sim q}[D(X,\mu)]$$
> follows from an intuitive property of the KL divergence. We explored extending this argument from KL divergence to general f-divergences; however, the broader class of functions does not, in general, satisfy the same monotonicity property as the function f associated with KL, and therefore the corresponding conclusion no longer holds in such a clear and interpretable form. We appreciate your suggestion and plan to leave a more general treatment of f-divergences to future work.

---

> ### Author Response · Authors · 2025-11-21
>
> ## Weakness 3 & Question 2
> > … Under limited computational resources, how would the quality, diversity, and effectiveness of adversarial samples degrade if simpler or smaller PGMs were used? Would the method remain effective in low-resource settings?
>
> Thank you for your insightful comment. We agree that concept augmentation contributes substantially to improving diversity and performance. However, we wish to clarify that concept augmentation is not part of the core concept-based adversarial attack. It is merely an optional procedure used when constructing the concept distribution in cases where the available concept data lack sufficient diversity. In such situations, we employ modern generative models to enrich the dataset.
>
> Crucially, if a concept already comes with a sufficiently diverse dataset, or if a concept distribution is directly provided, no augmentation is required. In low-resource settings where concept augmentation is not feasible, and diversity is extremely limited (e.g., a single image), our method naturally degenerates to ProbAttack, which remains effective, as shown in our experiments.
>
> Thus, concept augmentation enhances performance but is not a dependency of the proposed attack method.
>
> ## Question 3
> > … Please discuss how to better balance this trade-off between transferability and quality, and explain the specific reasons behind the drop in transferability for larger M
>
> In the conservative strategy shown in Table 5, we select the sample with the lowest softmax probability among the successful adversarial examples. As the sample size $M$ increases, we are more likely to find an adversarial example that barely fools the white-box classifier while also having the smallest softmax probability, that is, the sample farthest from the center of the red distribution $p_\text{vic}^{(1)}$ and thus closer to the center of $p_\text{dis}$. As illustrated in Figure 6 of the revised appendix, if the white-box classifier corresponds to the red distribution $p_\text{vic}^{(1)}$ and the transfer target corresponds to the green distribution $p_\text{vic}^{(2)}$, then selecting samples that lie farther away from the center of $p_\text{vic}^{(1)}$ will also push them farther from the center of $p_\text{vic}^{(2)}$. This naturally leads to a reduction in transferability.
>
>
> -----
>
> Once again, thank you for your participation. The clarity, theoretical depth, and overall quality of this paper have improved significantly, and this would not have been possible without your valuable suggestions.

---

### Official Review · Reviewer_ggXr · 2025-10-28

**Soundness:** 3
**Presentation:** 3
**Contribution:** 2
**Rating:** 4
**Confidence:** 4

**Summary:**

This paper introduces **concept-based adversarial attacks**, where a “concept” is modeled not as a single image but as a set of images or a fine-tuned generative model (e.g., diffusion). Adversarial examples are sampled from the overlap between the concept prior and the target classifier’s probability distribution, enabling identity-preserving attacks beyond norm-bounded perturbations. The authors provide theoretical justification via KL divergence analysis. They conduct extensive experiments on ImageNet, achieving **97.8% targeted white-box success**.  Perceptual quality is validated using CLIP similarity and no-reference IQA metrics such as MUSIQ, NIMA, and TReS.

**Strengths:**

1. The problem is well-motivated. The paper goes beyond traditional single-image or class-level attacks and instead enables identity-level, concept-aware adversarial generation that produces realistic and semantically consistent examples. Furthermore, I believe this approach could be valuable beyond adversarial attacks. It may help future work probe model hallucination and understand the semantic priors that models rely on.
2. The method is built on a clear probabilistic formulation rather than heuristic manipulation, making the approach principled and conceptually well-grounded.
3. The experiments include strong perceptual fidelity evaluation, using CLIP similarity and no-reference IQA metrics such as MUSIQ and TReS to assess visual realism.

**Weaknesses:**

1.  **Missing compute / FLOP parity.**
    The appendix briefly reports compute but does not provide a clear, quantitative comparison of **FLOPs / GPU-hours** between the proposed pipeline and the baselines. Please report wall-clock GPU-hours and/or FLOP counts per concept (training + sampling) for the proposed method and for each baseline. This will help readers judge whether performance gains are due to algorithmic novelty or to much greater compute and data budgets.

2.  **Experimental structure and choice of victim models.**
    The experimental section lacks a systematic structure for robustness testing. I request the following additions:


-   Report how attacks generated against ResNet-50 behave when transferred to an **adversarially-trained ResNet-50** under an L∞ threat model. This gives a clear point of comparison with standard robustness tests.

-   Include **white-box evaluations where the victim itself is adversarially trained** (i.e., generate attacks against a robustly trained model) to show whether the concept-guided sampling can break robust classifiers in white-box settings.

-   Broaden the set of victim models beyond standard classifiers. Evaluate against models trained with mainstream non-adversarial robustness techniques (e.g., CutMix, AugMix, models tuned for OOD/generalization). These are realistic victim models and better reflect deployment settings.


3.  **Missing strong imperceptible / transfer baselines.**
    The current baselines omit important, stronger transfer/imperceptible attacks. Please include comparisons to state-of-the-art transfer methods such as **VMI-FGSM** [ii] and **LGV** [i]. These methods represent a competitive, low-compute class of attacks and are necessary to understand the relative potency of the proposed generative approach.


[i]. LGV: Boosting Adversarial Example Transferability from Large Geometric Vicinity
[ii]. Enhancing the Transferability of Adversarial Attacks through Variance Tuning

**Questions:**

1. In Section 6, you state that prior work “treats a class (e.g., cat, dog, truck) as the concept, which cannot precisely capture an individual identity,” whereas your method represents a concept using a set of images or a probabilistic model. This suggests that your notion of ‘concept’ operates at an identity level rather than a class level. However, most related work in diffusion-based adversarial attacks defines ‘concept’ strictly at the class/category level. Can you please clarify your use of the term ‘concept’ more explicitly and explain how it differs from the class-level definition in existing work?

2. Please formalize your notion of transferability by giving a precise metric (for example top1 or top5 targeted success) and describing exactly how adversarial samples are generated and evaluated across source and target models. What am I precisely asking is that correctly classified samples by source model should not be used for transferability and that should reflect in the quantitative data presented under transferability.

3. How essential is the SDXL based augmentation step? Please explain why you need SDXL to build the concept dataset instead of using a large pretrained generative model trained on ImageNet or similar corpora, and include an ablation if possible.

4. How would a black box query attack operate in your framework and how does its expected strength compare to the transfer based evaluation you report?

---

> ### Author Response · Authors · 2025-11-24
>
> Thank you for your review of our work. Below is our response:
>
> ## Weaknesses
> ### Missing compute / FLOP parity.
>
> > Please report wall-clock GPU-hours and/or FLOP counts per concept (training + sampling) for the proposed method and for each baseline.
>
> Thank you for pointing this out. We agree that Appendix I in the original submission did not provide sufficiently detailed reporting of computing resources and only included the GPU time for our own method. In the revised version, we have added the full time breakdown for all methods evaluated in the paper.
>
> The table below summarizes the wall-clock time of each method on an NVIDIA H100 GPU, using ResNet-50 as the victim classifier (times may vary slightly when using different victim classifiers):
>
> | Task      | NCF | ACA | DiffAttack | ProbAttack | OURS |
> |-------------------------------------------|-----|-----|------------|------------|------|
> | Concept augmentation (per concept)        |  -  |  -  |     -      |     -      | 20m  |
> | Concept finetune (per concept)            |  -  |  -  |     -      |    20m     |  8h  |
> | Adversarial example generation (per image)| 30s | 20s |    12s     |    75s     | 75s  |
>
>
> We have incorporated this table into Appendix I of the revised manuscript.
>
> To ensure that the diffusion model learns a high-quality identity-level concept, we employ a long training schedule during the concept finetuning stage. When the concept dataset is sufficiently diverse (e.g., the 30 augmented images used in the main paper), the diffusion model remains stable and does not collapse even after many epochs. In contrast, in settings such as ProbAttack where finetuning is performed on a single image, excessive training easily leads to model collapse, and therefore the number of training steps must be carefully restricted. For this reason, the concept-finetuning time for ProbAttack is approximately 20 minutes, whereas our concept-based method requires around 8 hours.
>
> Across all concepts in our experiments, we verified that this 8-hour finetuning schedule works reliably. In practice, this time can be reduced to some extent; however, maintaining an identity-level concept is inherently subjective and difficult to quantify, making it challenging to specify a universally optimal finetuning duration.
>
>
> > This will help readers judge whether performance gains are due to algorithmic novelty or to much greater compute and data budgets.
>
> Your concern is very reasonable. We would like to clarify that although our method incurs substantially higher computational cost, it is also the only adversarial attack approach capable of leveraging this additional cost to preserve identity-level concepts (as discussed in our revised Related Work section). Existing baselines are fundamentally single-image attack methods, and even with more computation they cannot maintain identity-level semantics. Thus, while the comparison is inherently imperfect, our method is currently the only one for which increased training time can meaningfully translate into preserving an identity-level concept.

---

> ### Author Response · Authors · 2025-11-24
>
> ### Experimental structure and choice of victim models.
>
> > Report how attacks generated against ResNet-50 behave when transferred to an adversarially-trained ResNet-50 under an L∞ threat model.
>
> Thank you for the suggestion. In the revised version, we have added the transferability results on ResNet-50 Adv. to Table 10 in the appendix.
>
> > Include white-box evaluations where the victim itself is adversarially trained (i.e., generate attacks against a robustly trained model)
>
> Thank you for the suggestion! We conducted an additional experiment using ResNet-50 Adv as the white-box victim classifier, and the results are as follows:
>
> ASR:
>
> ||NCF|ACA|DiffAttack|ProbAttack|OURS (CONS)|OURS (AGGR)|
> |---|---|---|---|---|---|---|
> |**White-box**||||**Targeted-Top1**|||
> |ResNet-50 Adv|0.90|5.13|81.79|61.28|**94.23**|**94.23**|
> |**Transferability**||||||
> |VGG19|0.13|0.38|1.67|0.51|0.00|**2.31**|
> |ResNet 152|0.00|0.51|**2.56**|1.03|0.38|**2.56**|
> |DenseNet 161|0.00|0.38|2.31|0.77|0.38|**3.97**|
> |Inception V3|0.00|0.26|1.28|0.77|0.00|**2.18**|
> |EfficientNet B7|0.00|0.38|0.51|0.00|0.26|**1.67**|
> |**Adversarial Defence**||||||
> |Inception V3 Adv|0.00|0.51|1.67|0.51|0.00|**1.79**|
> |EfficientNet B7 Adv|0.00|0.90|1.15|0.64|0.51|**2.56**|
> |Ensemble IncRes V2|0.00|0.51|0.64|1.15|0.00|**2.31**|
> |**White-box**||||**Targeted-Top5**|||
> |ResNet-50 Adv|2.82|9.62|89.74|75.00|**98.08**|**98.08**|
> |**Transferability**||||||
> |VGG19|1.03|2.31|5.77|3.33|2.82|**6.15**|
> |ResNet 152|1.15|2.69|**11.28**|4.49|3.85|**11.79**|
> |DenseNet 161|1.15|2.82|10.00|5.38|5.26|**15.38**|
> |Inception V3|0.77|1.92|4.36|3.46|1.79|**6.67**|
> |EfficientNet B7|1.15|2.31|2.56|1.92|1.79|**5.51**|
> |**Adversarial Defence**||||||
> |Inception V3 Adv|0.90|1.92|5.13|3.46|1.41|**6.15**|
> |EfficientNet B7 Adv|0.26|1.79|3.33|3.59|2.56|**10.00**|
> |Ensemble IncRes V2|0.64|1.92|4.36|3.08|1.15|**7.95**|
> |**White-box**||||**Targeted-Top10**|||
> |ResNet-50 Adv|3.85|11.67|92.95|77.56|**98.97**|**98.97**|
> |**Transferability**||||||
> |VGG19|2.18|3.72|8.97|4.62|4.36|**10.77**|
> |ResNet 152|2.05|4.87|**18.97**|7.44|8.08|**19.49**|
> |DenseNet 161|1.92|5.00|16.28|7.95|11.15|**25.13**|
> |Inception V3|1.03|3.21|6.92|4.62|3.08|**8.72**|
> |EfficientNet B7|1.92|3.85|5.64|4.36|4.87|**8.72**|
> |**Adversarial Defence**||||||
> |Inception V3 Adv|1.41|2.69|8.21|5.38|3.59|**10.51**|
> |EfficientNet B7 Adv|0.38|3.08|5.00|5.38|3.85|**14.74**|
> |Ensemble IncRes V2|1.03|3.08|6.15|4.74|2.82|**11.41**|
>
> Image quality:
>
> |                      | Clean    | NCF      | ACA      | DiffAttack | ProbAttack | OURS (CONS) | OURS (AGGR) |
> |----------------------|----------|----------|----------|------------|------------|-------------|-------------|
> | **Similarity**       |          |          |          |            |            |             |             |
> | ↑ Avg. Clip Score    | 1.0      | **0.8556** | 0.7485   | 0.7877     | 0.8367     | 0.8059      | 0.7945      |
> | **Image Quality**    |          |          |          |            |            |             |             |
> | ↑ HyperIQA           | 0.7255   | 0.4854   | 0.6221   | 0.5297     | 0.6617     | **0.6875**  | 0.6750      |
> | ↑ DBCNN              | 0.6956   | 0.4908   | 0.5718   | 0.5070     | 0.6078     | **0.6473**  | 0.6342      |
> | ↑ ARNIQA             | 0.7667   | 0.5764   | 0.6654   | 0.6683     | 0.6932     | **0.7231**  | 0.7070      |
> | ↑ MUSIQ-AVA          | 4.3760   | 3.7435   | 4.2637   | 4.0130     | 4.2223     | **4.5031**  | 4.4966      |
> | ↑ NIMA-AVA           | 4.5595   | 3.7453   | 4.4342   | 4.0113     | 4.4676     | **4.7399**  | 4.7351      |
> | ↑ MUSIQ-KonIQ        | 65.0549  | 49.3740  | 57.0028  | 51.7202    | 58.1513    | **63.0946** | 61.3527     |
> | ↑ TReS               | 93.2127  | 62.8961  | 82.7340  | 72.1034    | 84.3081    | **90.4438** | 88.0786     |
>
> These results have been added to the appendix of the revised version.
>
> From the results, we observe that when using ResNet-50 Adv as the victim classifier, the white-box attack performance decreases compared to using the standard ResNet-50, while the transferability increases. This behavior is expected: adversarially trained classifiers tend to develop stronger semantic understanding, causing the victim distribution to concentrate more heavily on semantic features. As a result, adversarial examples drawn from the intersection of distributions tend to contain stronger target-class semantics, which naturally leads to higher transferability. This observation is consistent with our discussion in Appendix H: Additional Comments on Transferability.

---

> ### Author Response · Authors · 2025-11-24
>
> ### Experimental structure and choice of victim models. (Continued)
>
> > Broaden the set of victim models beyond standard classifiers. Evaluate against models trained with mainstream non-adversarial robustness techniques (e.g., CutMix, AugMix, models tuned for OOD/generalization). These are realistic victim models and better reflect deployment settings.
>
> Thank you for the suggestion. We would like to clarify that the victim models used in our experiments are drawn directly from the official pretrained checkpoints provided by PyTorch or timm library, which are commonly deployed in real-world production settings. Moreover, techniques such as CutMix, AugMix, and other mainstream non-adversarial robustness augmentations have already been integrated into the official PyTorch training pipeline and are incorporated into the training of these widely used classifier checkpoints. For details, please refer to the official PyTorch training script: https://github.com/pytorch/vision/blob/main/references/classification/train.py
>
> ### Missing strong imperceptible / transfer baselines.
>
> Thank you for the suggestion. We have cited the papers you mentioned and incorporated a discussion into Appendix H: Additional Comments on Transferability.
>
> As I understand your point, the methods you referred to are still Lp-norm bounded, untargeted attacks, which are not directly comparable to unrestricted adversarial attacks. For example, even color-based unrestricted attacks such as NCF already outperform Lp-based attacks in the relatively easier untargeted setting. However, under the much more challenging targeted setting used in our paper, NCF becomes one of the weakest baselines. This further highlights the fundamental differences between Lp-bounded and unrestricted attack paradigms.
>
> ## Questions
>
> ### Question 1
>
> > … Can you please clarify your use of the term ‘concept’ more explicitly
>
> Thank you for pointing this out. We agree that the definition of concept in the original submission was not sufficiently clear. In the revised version, we have clarified the definition more explicitly. Please see Section 3 for the updated explanation.
>
> > … and explain how it differs from the class-level definition in existing work?
>
> Please refer to the beginning of Section 3.1 in the revised manuscript. The following sentence provides a clear explanation of the distinction between identity-level concepts and class-level concepts:
>
> “A concept is inherently abstract and subjective: it may refer to a specific physical object (e.g., a rubber duck), a particular identity such as the long-eared corgi puppy with a lighter left cheek shown in Figure 1, or a broader class such as ``corgi,'' regardless of age, size, or specific attributes.”
>
> In our framework, an identity-level concept corresponds to a specific individual (e.g., one particular corgi), whereas a class-level concept refers to an entire category (e.g., all corgis) regardless of variations in size, age, or appearance.
>
> ### Question 2
>
> > Please formalize your notion of transferability by giving a precise metric (for example top1 or top5 targeted success) and describing exactly how adversarial samples are generated and evaluated across source and target models.
>
> First, we generate N adversarial examples using the white-box classifier and the chosen target label.
>
> We then evaluate transferability using the attack success rate (ASR). The procedure is straightforward: given the adversarial examples ${a_1, a_2, \ldots, a_N}$ and their corresponding target labels ${y_1, \ldots, y_N}$, the ASR is computed as:
>
> $$\text{ASR} = \frac{1}{N} \sum_{i=1}^{N} \mathbf{1}\big( y_i \in \text{top}_k(f(a_i)) \big),$$
>
> where $f$ denotes the black-box classifier and $\text{top}_k(\cdot)$ returns the set of top-$k$ predicted classes.
>
>
> > What am I precisely asking is that correctly classified samples by source model should not be used for transferability and that should reflect in the quantitative data presented under transferability.
>
> We understand your concern; however, adopting this evaluation protocol wouldinflate the transferability of methods whose white-box success rate is extremely low. For example, NCF achieves less than 5% white-box success - under your suggested calculation, its transferability would appear disproportionately high.
>
> Our transferability computation follows the same evaluation protocol used in the official implementations of [DiffAttack](https://github.com/WindVChen/DiffAttack/tree/main), [ACA](https://github.com/Omenzychen/Adversarial_Content_Attack), and [NCF](https://github.com/VL-Group/Natural-Color-Fool), ensuring a fair and consistent comparison across all baselines.

---

> ### Author Response · Authors · 2025-11-24
>
> ### Question 3
>
> > How essential is the SDXL based augmentation step?
>
> Please allow us to refer to our response to Reviewer 2if2, Question 2: “We agree that concept augmentation contributes substantially to improving diversity and performance. However, we wish to clarify that concept augmentation is not part of the core concept-based adversarial attack. It is merely an optional procedure used when constructing the concept distribution in cases where the available concept data lack sufficient diversity. In such situations, we employ modern generative models to enrich the dataset.
>
> Crucially, if a concept already comes with a sufficiently diverse dataset, or if a concept distribution is directly provided, no augmentation is required. In low-resource settings where concept augmentation is not feasible, and diversity is extremely limited (e.g., a single image), our method naturally degenerates to ProbAttack, which remains effective, as shown in our experiments.
>
> Thus, concept augmentation enhances performance but is not a dependency of the proposed attack method.”
>
>
> > Please explain why you need SDXL to build the concept dataset instead of using a large pretrained generative model trained on ImageNet or similar corpora, and include an ablation if possible.
>
> This is related to your Question 1, the SDXL-based concept we construct corresponds to an identity-level concept, whereas ImageNet provides only class-level concepts.
>
>
> ### Question 4
>
> > How would a black box query attack operate in your framework and how does its expected strength compare to the transfer based evaluation you report?
>
> Thank you for raising this question. We acknowledge that our current framework does not yet support black-box query attacks. We consider this an important direction for future work and plan to explore it in subsequent research.
>
> -----
>
> Because the additional experiments required substantial computation time, our response is slightly later than the 20th. We apologize for any inconvenience this may have caused. Once again, thank you for your suggestions, your feedback has greatly improved the rigor and quality of our experiments.

---

### Official Review · Reviewer_av86 · 2025-11-01

**Soundness:** 2
**Presentation:** 3
**Contribution:** 3
**Rating:** 6
**Confidence:** 4

**Summary:**

The paper extends the probablistic framework for adversarial examples presented in Zhang et al. (Neurips 2024) to an unrestricted, concept based setting. The basic idea relies on replacing the image-based specific distance (defined in Zhang et al.). with a generalized concept-based distance. The authors surrogate "concept" with a probabilistic model (diffusion models in this case) by finetuning a pretrained diffusion model on a set of images, and then sample adversarial candidates. They provide two sampling strategies as well as theoretical intuition for the framework. They further showcase empirical results to support their claim that concept priors reduce to an empirical KL gap and produce high quality adversarial examples when compared to other unrestricted approaches like DiffAttack.

**Strengths:**

1. The presented framework is a clean and well motivated generalization of the probabilistic framework presented in Zhang et al. The idea of moving away from an image-centric distance distribution to a concept-prior through the use of finetuned diffusion models is inspired.

2. Empirical performance of the given approach is encouraging, and the results support the authors' claims of better, and more semantically meaningful adversarial examples as compared to methods like DiffAttack.

3. Implementing the concept-prior using finetuned SDXL models is an interesting approach, and can possibly be generalized to any set of images with common features. This opens up a new mechanism of measuring vulnerabilities of classifiers.

**Weaknesses:**

1. The theoretical contributions are mostly incremental with both Thm.1 and 2 being straightforward algebra. While supportive of the presented conceptual framework, it does not really provide any additional insight on how the approach can be further optimized or adapted to specific PGMs like diffusion models.

2. The transferability results are extremely low.This suggests very low overlap between $p_{vic}$ and $p_{dis}$ which is a bit counterintuitive given the strong performance of these classifiers on such tasks. Shouldn't a concept based prior have a much stronger overlap?  Do the authors have any hypothesis about this? The top-100 results in the appendix are also uninformative given the weak constraints of the experiment and should be removed.

3. The models presented in this work are also older. How does this attack fair with newer architectures like CLIP, ConvNext etc?

**Questions:**

1. Please provide more details on the baseline comparisons for DiffAttack and others. Specifically, number of sampling steps, solvers used and any other hyperparameters.

---

> ### Author Response · Authors · 2025-11-22
>
> Thank you for your review of our work. Below is our response to the weaknesses and questions:
>
> ## About the Theorems
>
> > … While supportive of the presented conceptual framework, it does not really provide any additional insight on how the approach can be further optimized or adapted to specific PGMs like diffusion models.
>
> Thank you for pointing this out. We agree that the Gaussian assumption in the original submission’s Thm. 1 was overly idealized and did not provide meaningful insight into how modern PGMs such as EBMs or diffusion models benefit from the theory. Following your suggestion, we realized that the Gaussian assumption can be naturally generalized to a Gibbs distribution. Accordingly, in the revised version we have updated Thm. 1 and its proof in the appendix.
>
> This generalization significantly improves the theorem’s relevance to contemporary PGMs: Energy-Based Models (EBMs) explicitly model probability distributions in the form of a Gibbs distribution, while diffusion models can be interpreted as implicitly learning the energy function of a Gibbs distribution. This substantially enhances the applicability of Thm. 1 to modern generative modeling frameworks.
>
> Thm. 2 does not assume any specific form for  $p_\text{dis}$, and therefore applies directly to modern PGMs. Indeed, Thm. 2 serves as the theoretical basis for the empirical study presented in Section 5.3.
>
> ## About Transferability
>
> > … which is a bit counterintuitive given the strong performance of these classifiers on such tasks. Shouldn't a concept based prior have a much stronger overlap? Do the authors have any hypothesis about this?
>
> Thank you for raising this point. To address this question, please refer to the updated Appendix H: Additional Comments on Transferability in our revised version. There, we introduce a probabilistic perspective on transferability. As illustrated in Figure 6, a concept-based prior indeed yields a stronger overlap - but this overlap occurs with the white-box classifier’s victim distribution (the red distribution), not with the black-box classifier’s victim distribution (the green distribution).
> Consequently, adversarial examples sampled from the intersection of the red and blue distributions do not achieve high probability density under the green distribution. This explains why transferability increases with a concept-based prior, yet remains limited in absolute terms.
>
> > The top-100 results in the appendix are also uninformative given the weak constraints of the experiment and should be removed.
>
> Thank you for the suggestion. We have removed the top-100 results from the appendix.

---

> ### Author Response · Authors · 2025-11-22
>
> ## Additional Experiments
>
> > The models presented in this work are also older. How does this attack fair with newer architectures like CLIP, ConvNext etc?
>
> Thank you for the suggestion. Since CLIP is not a standard classifier, we additionally conducted experiments using ConvNeXt as the white-box model. The results are shown below:
>
> ASR:
>
> | | NCF | ACA | DiffAttack | ProbAttack | OURS (CONS) | OURS (AGGR) |
> |---|---|---|---|---|---|---|
> | **White-box** | | | **Targeted-Top1** | | | |
> | ConvNext | 1.03 | 5.38 | 83.59 | 60.38 | **94.74** | **94.74** |
> | **Transferability** | | | | | | |
> | VGG19 | 0.00 | 0.26 | **1.28** | 0.26 | 0.00 | 1.15 |
> | ResNet 152 | 0.00 | 0.38 | 1.54 | 0.51 | 0.13 | **1.67** |
> | DenseNet 161 | 0.00 | 0.26 | 1.54 | 0.51 | 0.13 | **2.31** |
> | Inception V3 | 0.13 | 0.26 | 0.77 | 0.38 | 0.00 | **1.28** |
> | EfficientNet B7 | 0.00 | 0.26 | 0.51 | 0.00 | 0.13 | **1.03** |
> | **Adversarial Defence** | | | | | | |
> | Inception V3 Adv | 0.00 | 0.26 | 0.90 | 0.26 | 0.00 | **1.03** |
> | EfficientNet B7 Adv | 0.00 | 0.51 | 0.77 | 0.38 | 0.38 | **1.28** |
> | Ensemble IncRes V2 | 0.00 | 0.26 | 0.51 | 0.51 | 0.00 | **1.28** |
> | **White-box** | | | **Targeted-Top5** | | | |
> | ConvNext | 2.95 | 9.36 | 91.15 | 78.85 | **98.59** | **98.59** |
> | **Transferability-Top5** | | | | | | |
> | VGG19 | 1.15 | 1.54 | 3.46 | 2.05 | 1.79 | **4.10** |
> | ResNet 152 | 1.15 | 1.79 | 6.92 | 2.69 | 2.44 | **7.56** |
> | DenseNet 161 | 1.15 | 1.92 | 6.67 | 3.33 | 3.21 | **9.62** |
> | Inception V3 | 0.90 | 1.41 | 3.08 | 2.18 | 1.15 | **4.23** |
> | EfficientNet B7 | 1.28 | 1.79 | 2.05 | 1.41 | 1.28 | **3.72** |
> | **Adversarial Defence-Top5** | | | | | | |
> | Inception V3 Adv | 0.90 | 1.41 | 3.21 | 1.67 | 0.90 | **3.59** |
> | EfficientNet B7 Adv | 0.26 | 1.28 | 2.44 | 2.05 | 1.54 | **5.51** |
> | Ensemble IncRes V2 | 0.64 | 1.28 | 2.82 | 1.67 | 0.77 | **4.49** |
> | **White-box** | | | **Targeted-Top10** | | | |
> | ConvNext | 3.97 | 11.67 | 93.97 | 81.54 | **98.72** | **98.72** |
> | **Transferability-Top10** | | | | | | |
> | VGG19 | 2.44 | 2.69 | **6.67** | 2.95 | 3.08 | 6.15 |
> | ResNet 152 | 2.05 | 3.08 | 11.92 | 4.49 | 5.13 | **12.82** |
> | DenseNet 161 | 1.92 | 3.46 | 10.90 | 5.13 | 7.18 | **16.41** |
> | Inception V3 | 1.28 | 2.18 | 5.00 | 2.82 | 2.18 | **6.15** |
> | EfficientNet B7 | 2.05 | 2.82 | 4.36 | 2.95 | 3.46 | **6.28** |
> | **Adversarial Defence-Top10** | | | | | | |
> | Inception V3 Adv | 1.41 | 1.92 | 5.13 | 3.08 | 2.18 | **6.15** |
> | EfficientNet B7 Adv | 0.38 | 2.18 | 3.72 | 3.21 | 2.69 | **8.21** |
> | Ensemble IncRes V2 | 1.15 | 2.18 | 4.23 | 2.82 | 2.05 | **6.54** |
>
>
> Image quality:
>
> |  | Clean | NCF | ACA | DiffAttack | ProbAttack | OURS (CONS) | OURS (AGGR) |
> |---|---|---|---|---|---|---|---|
> | **Similarity** |
> | ↑ Avg. Clip Score | 1.0 | **0.8621** | 0.7542 | 0.7964 | 0.8467 | 0.8139 | 0.8011 |
> | **Image Quality** |
> | ↑ HyperIQA | 0.7255 | 0.4928 | 0.6281 | 0.5392 | 0.6703 | **0.6933** | 0.6810 |
> | ↑ DBCNN | 0.6956 | 0.4951 | 0.5784 | 0.5179 | 0.6201 | **0.6531** | 0.6405 |
> | ↑ ARNIQA | 0.7667 | 0.5835 | 0.6710 | 0.6801 | 0.7066 | **0.7294** | 0.7118 |
> | ↑ MUSIQ-AVA | 4.3760 | 3.7511 | 4.2814 | 4.0206 | 4.2314 | **4.5116** | 4.5030 |
> | ↑ NIMA-AVA | 4.5595 | 3.7524 | 4.4433 | 4.0215 | 4.4830 | 4.7464 | **4.7492** |
> | ↑ MUSIQ-KonIQ | 65.0549 | 49.3812 | 57.0114 | 51.7324 | 58.9012 | **63.1025** | 61.3571 |
> | ↑ TReS | 93.2127 | 62.9011 | 82.7410 | 72.1109 | 85.2442 | **90.9211** | 88.7724 |
>
> These results have been added to the appendix of the revised version.
>
> ## More Details
>
> > Please provide more details on the baseline comparisons for DiffAttack and others.
>
> Thank you for pointing this out. Below we provide more details on the baseline comparisons:
>
> | Method | Sampler / Solver | # Sampling Steps | # Attack Steps | Step Size | Notes |
> |---|---|---|---|---|---|
> | DiffAttack | DDIM | 50 | 30 | 0.01 | - |
> | ACA | DDIM | 50 | 10 | 0.04 | - |
> | NCF | - | - | 15 | 0.013 | # Color Sampling = 10 |
> | ProbAttack | DDPM | 250 | - | - | \|C_ori\| = 1, M = 10, c = 30. |
> | Concept-based | DDPM | 250 | - | - | \|C_ori\| = 30, M = 10, c = 30. |
>
> We list the hyperparameter settings for all compared methods in the table above, all other parameters are at default. Note that DiffAttack/ACA and ProbAttack/Concept-based Attack rely on fundamentally different attack mechanisms: the former are based on DDIM and operate in the latent space, while the latter directly sample from $p_\text{adv}$. As a result, the number of sampling steps is not directly comparable across these methods, and the latter do not require any additional attack steps.
>
> We have incorporated the above table and the corresponding discussion into Appendix P of the revised version.
>
> -----
>
> Once again, thank you for your valuable suggestions. Your feedback has greatly strengthened the paper - from the theoretical formulation to the conceptual insights and the clarity of the experimental descriptions.

---

### Official Review · Reviewer_vSZj · 2025-11-01

**Soundness:** 3
**Presentation:** 4
**Contribution:** 3
**Rating:** 8
**Confidence:** 4

**Summary:**

The paper introduces Concept-Based Adversarial Attacks, a novel framework that extends traditional image-based adversarial attacks by operating not on a single image but on an entire concept represented either by a set of images or a probabilistic generative model (e.g., a fine-tuned diffusion model). The method adopts a probabilistic perspective, where adversarial examples are sampled from the product of a “victim” distribution (encouraging misclassification) and a “distance” distribution centered on the concept rather than a single image. This allows generation of diverse adversarial examples that preserve the underlying identity or object (e.g., a specific dog) while varying pose, background, or viewpoint to fool classifiers. The approach is theoretically grounded in KL divergence analysis and empirically validated on ImageNet, showing higher attack success rates and better concept preservation than prior methods.

**Strengths:**

The main contributions (and strengths) of the paper are:

* Novel formulation: First work to define adversarial distance at the concept level rather than per-image, enabling more semantically meaningful attacks.

* Strong empirical results: Achieves state-of-the-art targeted attack success rates (e.g., 97.82% white-box on ResNet-50) while better preserving concept identity (validated via user studies and CLIP scores).

* Theoretical justification: Provides analysis showing that expanding the distance distribution to a concept reduces KL divergence between victim and distance distributions, increasing overlap and attack efficacy.

* Practical pipeline: Integrates modern generative models (Stable Diffusion XL + LoRA) and LLMs (GPT-4o) for concept augmentation, enhancing diversity and realism.

**Weaknesses:**

Computational cost: The proposed approach requires fine-tuning generative models per concept, which is time-consuming (≈8 hours/concept) and limits scalability.

*Limited transferability. While the experimental results show strong performance on white-box attacks, black-box transfer success remains low (though better than baselines), especially under strict top-1 metrics.

* Concept definition ambiguity. The proposed Relies on user-provided image sets or fine-tuned models to define a “concept,” which may vary in quality or scope and lacks a formal mathematical definition.

* Ethical risk. The proposed approach enables highly realistic, identity-preserving attacks that could evade content moderation systems, raising safety concerns. While the authors discuss mitigations, it’s unclear how these would deter malicious actors. I recognize the importance of publishing sensitive research to foster community dialogue, but is merely discouraging harmful use sufficient when the technique itself carries demonstrable ethical hazards?

**Questions:**

Please refer to the section above.

---

> ### Author Response · Authors · 2025-11-21
>
> Thank you for your review of our work. Below is our response to the identified weaknesses:
>
> ## Computational cost
> > Computational cost: The proposed approach requires fine-tuning generative models per concept, which is time-consuming (≈8 hours/concept) and limits scalability.
>
> Yes, in our paper, fine-tuning the diffusion model on a concept takes a relatively long time (~8 hours) because our base model is the OpenAI guided-diffusion implementation ([https://github.com/openai/guided-diffusion](https://github.com/openai/guided-diffusion)). This implementation is relatively old, lacks modern acceleration optimizations, and we perform full fine-tuning, which further increases the cost. We chose this model because it directly models $p(x)$ rather than $p(z)$, unlike popular models such as Stable Diffusion. This choice aligns with the theoretical formulation of our paper, whose primary goal is to introduce concept-based adversarial attacks as a new attack paradigm.
>
> Using models that parameterize $p(z)$ (Stable Diffusion) would be a natural engineering extension of our work. These models benefit from modern optimizations (e.g., efficient LoRA fine-tuning), making training significantly faster. However, in this paper we aimed to present a principled and theoretically grounded formulation of probabilistic adversarial attacks, so we deliberately used a $p(x)$-based model even though it is slower.
>
> ## Limited transferability
>
> > Limited transferability. While the experimental results show strong performance on white-box attacks, black-box transfer success remains low (though better than baselines), especially under strict top-1 metrics.
>
> Your observation is entirely accurate. Although our method achieves higher transferability compared with other comparable approaches, there remains a noticeable gap relative to methods explicitly optimized for transferability. We have strengthened our explanation and analysis in the revised version, specifically in Appendix H: Additional Comments on Transferability. In the updated discussion, we reinterpret transferability from a probabilistic perspective (see Figure 6), explain why our method still falls short in transferability, and outline possible directions for future exploration.
>
> ## Concept definition ambiguity
>
> > Concept definition ambiguity. The proposed Relies on user-provided image sets or fine-tuned models to define a “concept,” which may vary in quality or scope and lacks a formal mathematical definition.
>
> Thank you for pointing this out. We agree that defining a “concept’’ directly through user-provided image sets or fine-tuned models can introduce mathematical ambiguity. In the revised version, we have clarified this definition: mathematically, we represent a concept using a concept distribution, and this distribution is then approximated by a model or by a model trained on an image set. Please refer to the updated Section 3 in our revision for the details.
>
> ## Ethical risk
>
> > … it’s unclear how these would deter malicious actors. I recognize the importance of publishing sensitive research to foster community dialogue, but is merely discouraging harmful use sufficient when the technique itself carries demonstrable ethical hazards?
>
> We appreciate the reviewer’s concern and acknowledge the dual‑use risks inherent in concept‑conditioned attacks. Our intent is to rigorously characterise this underexamined vulnerability so that risks can be assessed with clarity rather than obscurity.
>
> Importantly, as outlined in Appendix L, our work studies a single theoretical attack mechanism, whereas real-world deployments employ layered, heterogeneous defenses. These include adversarial training, provenance- and watermark-based detection, and AI-generated content detection. Thus, the practical safeguards available in real systems extend far beyond simple discouragement.
>
> The crux of our position remains: we believe transparent, carefully contextualised reporting is the most responsible path to catalysing safeguards that will prevent abusive adversarial attack scenarios from emerging in real‑world deployments.
>
> -----
>
> Please allow us to express our gratitude for your review once again. Following your suggestions, we restructured the definition of concept in the paper, conducted a deeper analysis of transferability, and introduced a probabilistic perspective to explain it. Without your insightful feedback, the paper would not have reached its current level of completeness.

---

### Author Response · Authors · 2025-11-27

## **Revision Summary**

Since the discussion phase allows paper revisions, we summarize the key updates as follows:

1. **Clearer definition of concept.**
   Following suggestions from reviewers **vSZj** and **ggXr**, we refined the definition of concept.
   Previously, a concept was described loosely as “a model or a set of images,” which lacked clarity.
   In the revised version, we formally represent a concept using a **concept distribution**, which can be instantiated either by a probabilistic model conditioned on specified attributes, or by a probabilistic model trained on a set of images - consistent with the common assumption of an underlying true distribution $p_{\text{true}}$ in machine learning.
   Related revisions were made in **Section 3.1**, **Section 3.2**, and other parts of the paper.

2. **Generalizing Theorem 1 from Gaussian to Gibbs distributions.**
   Based on comments from reviewers **av86** and **2if2**, we extended Theorem 1 from the Gaussian case to the more general **Gibbs distribution**.
   This significantly broadens the theorem’s applicability and naturally accommodates both **energy-based models** and **diffusion models**.
   The corresponding revisions were made in **Section 3.4** and **Appendix A**.

3. **Expanded related work and consolidated comparison table.**
   Per reviewer **2if2**’s request, we enhanced the related-work discussion in **Section 6** and added **Table 3**, which systematically summarizes recent unrestricted adversarial attack methods.
   We also highlight that our work is **the only unrestricted adversarial attack capable of operating at the identity-level concept**.

4. **Added a probabilistic explanation of transferability.**
   Following feedback from reviewers **vSZj**, **av86**, **ggXr**, and **2if2**, we added a detailed discussion of transferability in **Appendix H**, along with a new **Figure 6** illustrating transferability from a probabilistic perspective.
   To our knowledge, this is the **first probabilistic explanation** of transferability in unrestricted adversarial attacks.
   It clarifies why our method improves transferability relative to prior work, while also explaining why the absolute level of transferability remains inherently limited.

5. **Compute-resource discussion and time comparison.**
   In response to reviewer **ggXr**, we added a compute-resource discussion in **Appendix I** and reported GPU time comparisons for all methods in **Table 8**.
   Although our concept fine-tuning stage is more time-consuming, our method remains the only mathematically principled unrestricted adversarial attack that supports **identity-level** concepts.
   As discussed with reviewer **vSZj**, this paper prioritizes theoretical alignment and principled methodology over engineering optimizations. We believe substantial engineering acceleration is possible but beyond the scope of the current work; our focus is on implicitly constraining identity-level concepts through a distributional formulation, which is crucial for future progress in adversarial attacks.

6. **Added baseline hyperparameter settings.**
   Following reviewer **ggXr**, we included **Appendix P: Baseline Settings**, listing the hyperparameters for all compared methods.
   We also emphasize that DiffAttack/ACA and ProbAttack/Concept-based Attack rely on fundamentally different mechanisms: the former use DDIM in the latent space, while the latter directly sample from $p_{\text{adv}}$.
   Therefore, the number of sampling steps across methods is **not directly comparable**, and the latter do not require additional attack steps.

7. **Additional experiments using ConvNeXt and ResNet-50-Adv.**
   Based on requests from reviewers **av86** and **ggXr**, we added experiments using **ConvNeXt** and **ResNet-50-Adv** as white-box victim classifiers. Results are provided in **Table 13**, **Table 14**, **Table 18**, and **Table 19** in **Appendix Q**.

---

We reaffirm that all revisions focus on clarifying definitions, improving readability, broadening theoretical applicability (Gaussian to Gibbs), expanding experiments, and adding explanations requested by reviewers.
We **did not introduce new methods**, **did not alter any claims**, and **did not change the essence of the theory or experiments**.
All revisions are moderate, reviewer-driven, and aimed at improving clarity and completeness.

---

We would like to once again express our sincere gratitude to all reviewers and the AC for their thoughtful engagement—this paper would not have reached its current level of clarity and completeness without your guidance. We kindly ask whether you could provide us with additional feedback on whether the revised version still contains any unresolved issues.

---

### Author Response · Authors · 2025-12-03
**Discussion Summary**

Dear Area Chair,

Thank you for taking over the review process under the unexpected circumstances. Given the transition and the freeze of the discussion phase, We would like to provide a concise summary to support your assessment:

-----

**Novelty**

Our work introduces concept-based adversarial attacks, extending traditional adversarial perturbations from the level of a single image to that of a concept described by a distribution. Reviewers generally acknowledged this innovation.

* **vSZj** and **ggXr** commented that the notion of a concept should be more sharply defined. In response, we revised the manuscript to provide a clearer and more formal mathematical definition.

* **2if2** suggested that we more clearly distinguish our concept-based attack from existing unrestricted adversarial attacks. In our rebuttal, we provided a table demonstrating that our concept-based formulation is the most flexible and the only one supporting identity-level concepts. This table has been added to the revised manuscript.

**Absolute values in transferability metrics**

In the original submission, we clarified that our results focus on targeted adversarial attacks, which are substantially more challenging than untargeted ones. Under this setting, our method achieves stronger performance than comparable approaches.

For top-1 targeted transferability, the absolute numbers are lower than methods specifically optimized for transferability. This is expected: those methods explicitly generate features of the target class. We added an explanation from the perspective of probabilistic adversarial attacks in Appendix H of the revision.

The primary aim of our paper is to show that enlarging the range of the underlying concept (i.e., extending the distance distribution from around a single image to around a concept) can increase the intersection of $p_\text{dis}$ and $p_\text{vic}$ under the probabilistic view, producing higher-quality adversarial samples. Transferability improves as a result, but deliberately optimizing transferability is not the focus of our work. As noted in Appendix H, this direction remains open for further exploration.

**Computational cost**

**ggXr** and **vSZj** pointed out the relatively high computational cost associated with concept finetuning at the identity level. We acknowledge this and expanded Appendix I to discuss it in detail.

The goal of our paper is to introduce a new attack paradigm. Although our current prototype is computationally expensive, it is the only approach capable of performing identity-level concept-based attacks (so far), representing a genuine step from 0 to 1. Moreover, as noted in our response to **ggXr**, the implementation choices in our paper prioritize theoretical alignment rather than engineering optimization.

-----

The above summarizes the points we believe are most important in the discussion. For additional details, please refer to our point-by-point rebuttal and the Revision Summary.

Thank you again for your effort during this challenging period.

Best Regards

Authors of Submission 17950

---

### Meta-Review · Area_Chair_GMus · 2026-01-05

**Summary:**

Reviewers mainly highlighted:
- Confusions about the definition of "concepts"
- Lack of theoretical justification that the concept-based approach yields better adversarial examples, especially when realized in practice, e.g., using diffusion models.
- Limited improvement in attack transferability
- Other comments regarding the choice of baselines and positioning wrt generative approaches to adversarial attacks

Additional comments:
- The keyword "concept" may create confusion with how it's used in explainability studies; see [1].  This is not helped by the lack of an actual definition of what a concept is (L163), which ends up being a "distribution" to be adapted based on the use case.
- Note that the introduction already highlights transferability as a motivation for unrestricted adversarial attacks, including the proposed approach.  It would help to highlight the limited transferability improvements achieved by the proposed approach as part of this discussion, or possibly restructure the introduction to make this more efficient.

[1] Poeta, Eleonora, Gabriele Ciravegna, Eliana Pastor, Tania Cerquitelli, and Elena Baralis. "Concept-based explainable artificial intelligence: A survey." ACM Computing Surveys (2023).

**Reviewer Concerns:**

The authors provided comprehensive rebuttals of seemingly all reviewer comments, already reflected in the revised manuscript.

That said, there remain a few suboptimal aspects of the submission, due to the way it was put together in the first place and followed by various patches to address issues raised by reviewers.  That said, I think the study is valuable to the community as is and doesn't warrant additional review cycles.  Hence, I'm recommending acceptance.

**Reviewer Scores:**

Based on the comprehensive and high-quality responses provided by the authors, I would expect all reviewers to receive the rebuttal favorably raising their scores.

Starting with 8/6/4/4 -- I believe author rebuttals would be sufficient for a final score closer to 7.

---

### Decision · Program_Chairs · 2026-01-26

Accept (Poster)